# Jet stream position explains regional anomalies in European beech forest productivity and tree growth

Isabel Dorado-Liñán [1✉], Blanca Ayarzagüena [2], Flurin Babst[3,4], Guobao Xu [4,5], Luis Gil[1], Giovanna Battipaglia [6], Allan Buras[7], Vojtěch Čada [8], J. Julio Camarero [9], Liam Cavin[10], Hugues Claessens[11], Igor Drobyshev [12], Balázs Garamszegi [13], Michael Grabner[14], Andrew Hacket-Pain [15], Claudia Hartl [16], Andrea Hevia [17], Pavel Janda[8], Alistair S. Jump[10], Marko Kazimirovic [18], Srdjan Keren [19], Juergen Kreyling [20], Alexander Land [21,22], Nicolas Latte [11], Tom Levanič [23,24], Ernst van der Maaten [25], Marieke van der Maaten-Theunissen [25], Elisabet Martínez-Sancho [26], Annette Menzel[27,28], Martin Mikoláš[8], Renzo Motta[29], Lena Muffler[30], Paola Nola [31], Momchil Panayotov[32], Any Mary Petritan [33], Ion Catalin Petritan[34], Ionel Popa [33,35], Peter Prislan [23], Catalin-Constantin Roibu [36], Miloš Rydval [8], Raul Sánchez-Salguero [37], Tobias Scharnweber [20], Branko Stajić [18], Miroslav Svoboda [8], Willy Tegel [38], Marius Teodosiu [39], Elvin Toromani[40], Volodymyr Trotsiuk [8,26], Daniel-Ond Turcu[33], Robert Weigel [30], Martin Wilmking [20], Christian Zang [7,41], Tzvetan Zlatanov [42] & Valerie Trouet [4]

The mechanistic pathways connecting ocean-atmosphere variability and terrestrial productivity are well-established theoretically, but remain challenging to quantify empirically. Such quantification will greatly improve the assessment and prediction of changes in terrestrial carbon sequestration in response to dynamically induced climatic extremes. The jet stream latitude (JSL) over the North Atlantic-European domain provides a synthetic and robust physical framework that integrates climate variability not accounted for by atmospheric circulation patterns alone. Surface climate impacts of north-south summer JSL displacements are not uniform across Europe, but rather create a northwestern-southeastern dipole in forest productivity and radial-growth anomalies. Summer JSL variability over the eastern North Atlantic-European domain (5-40E) exerts the strongest impact on European beech, inducing anomalies of up to 30% in modelled gross primary productivity and 50% in radial tree growth. The net effects of JSL movements on terrestrial carbon fluxes depend on forest density, carbon stocks, and productivity imbalances across biogeographic regions.

A full list of author affiliations appears at the end of the paper.

Jet stream variability orchestrates weather patterns and extremes at the Earth's surface on daily to seasonal time-scales[1]. Variability is most pronounced, and most often studied, in wintertime, when the jet stream is at its strongest[2]. The mid-latitude jet stream is weaker in summer, but can nevertheless also be an important dynamic driver of surface climate variability[3,4] and of extreme weather[5].

In Europe, summer weather is dynamically driven by jet stream latitude (JSL) over the North Atlantic–European domain, which is linked to the Atlantic storm tracks and the occurrence of persistent and strong anticyclonic anomalies that disrupt the westerly airflow (atmospheric blocking)[6,7]. Such atmospheric blocking can result in summer heatwaves and drought that may affect large areas[8].

A northern summer JSL displacement over the North Atlantic is typically associated with anomalously warm and dry weather in northwestern Europe, but cool and wet weather over southeastern Europe. The reverse pattern occurs during southern summer JSL positions, resulting in cold and wet anomalies over the British Isles and increased odds of heatwaves and drought over the Balkans[4,9].

In recent decades, the latitudinal variability of the North Atlantic jet has increased[4,10], resulting in an increasing number of mid-latitude extreme weather events[10,11]. The impact of such increased JSL variability as a dynamic driver of summer weather extremes can be further amplified by the synergic effect of other atmospheric anomalies or environmental hazards[12,13], leading to large-scale biosphere disturbances[14,15]. Reductions in ecosystem productivity as a consequence of extreme weather events can decrease both regional ecosystem carbon uptake and sequestration[16,17]. In this context, dynamically driven climate extremes tend to influence large geographic areas[18] and thus have substantial ecological and socio-economic impacts. For instance, two of the most extreme heatwaves in Eurasia in 2003 and 2010 were caused by persistent atmospheric blocking, linked to a persistent North Atlantic–European JSL anomaly[8,19]. These two heatwaves resulted in reductions in European ecosystem gross primary production (GPP) of 20% and 50%, respectively[20,21]. The impacts of these heatwaves can be further amplified by preceding climate conditions, such as the continental-scale spring soil-moisture deficit in 2003 that fed back into the climate system[22]. Projected increases in heatwave frequency under future anthropogenic warming[23] may thus compromise the increasing carbon storage trajectory observed in temperate European forests over recent decades[24,25].

Problematically, we lack a quantitative perspective on the dynamic drivers of European summer climate extremes, and particularly JSL variability, in relation to forest productivity. Here, we assess and quantify the physical coupling between summer North Atlantic–European JSL variability and anomalies in temperate European beech (Fagus sylvatica L.) radial growth and productivity over Europe. For this purpose, we first define the main modes of summer JSL variability over the North Atlantic–European realm and their relation with anomalous atmospheric circulation structures using NCEP/NCAR reanalysis data[26]. We then relate the main modes of summer JSL variability over the North Atlantic–European domain to radial tree growth using a unique network of 344 European beech tree-ring width (TRW) chronologies that reflect variable carbon allocation to woody biomass[27]. In addition, we derive ecosystem carbon uptake (i.e., GPP) variability for the locations of the tree-ring sites as the TRW chronologies using a state-of-the-art ensemble of Dynamic Global Vegetation Models (DGVMs). Our focus on extremes allows us to diagnose the synoptic-scale configurations and climatic fluctuations that trigger the most substantial carbon anomalies across European temperate forests.

## Results

**Dominant JSL modes and climate extremes over Europe**. We identified two main principal component JSL modes (the primary mode jslPC1 and the secondary mode jslPC2) that together explain 56% of July–August JSL variability over the North Atlantic–European domain (Supplementary Figure 2). The two modes differ primarily in the longitudinal band where the meridional displacement of the jet stream occurs, and this difference is most pronounced during jslPC1 and jslPC2 extremes (i.e., 1st and 9th deciles of PC scores; D10 and D90; Fig. 1a, d, g, j). The composite maps of the identified extreme years show that the primary mode jslPC1, which explains almost 30% of the variability, is characterized by latitudinal JSL movements in the western part of the North Atlantic–European domain (ca. 30W-10E), the typical exit region of the North Atlantic jet stream (Fig. 1a, d). The extremes of the primary mode (jslPC1_D90 and jslPC1_D10) therefore represent southern (jslPC1_D90; hereafter named "southwestern JSL") and northern (jslPC1_D10; "northwestern JSL") JSL anomalies in the western part of the North Atlantic–European domain. Extremes of this mode are exemplified by the summer droughts of 1969 and 1983 (D10) that affected northern and central Europe and the summer droughts of 1952 and 2000 (D90) that affected southern and eastern Europe[28]. The secondary summer JSL mode (jslPC2) explains an additional 26% of the variability and is characterized by north-south JSL displacements over central-eastern Europe (5W–40E) (Fig. 1g, j). Extremes of the second mode show the strongest differences over the European continent, with jslPC2_D90 ("southeastern JSL") reflecting southward displacements of the JSL relative to the climatological mean and jslPC2_D10 ("northeastern JSL") showing northward displacements over the eastern portion of the domain (Fig. 1g, j). Among the extremes of this mode are the droughts of 1959 and 2005 (D90) that affected central and northern Europe and the summer drought of 1985 (D10) that mainly affected southern and southeastern Europe[28].

Similar to the weather regimes described for the preferred jet stream position in wintertime[29], the phases and extremes of the two main modes of summer JSL variability reflect dominant summer weather regimes over the North Atlantic–European domain[30]. Southward migrations of summer JSL (i.e., southwestern and southeastern; Figs. 1a, g) are driven by cyclonic regimes, whereas northern summer JSL positions (i.e., northwestern and northeastern; Figs. 1d, j) are driven by anticyclonic regimes (i.e., atmospheric blocking and ridges).

Southwestern JSL anomalies occur when a strong cyclone over western Europe deflects the prevailing westerly flow southward (Fig. 1a). The cyclone co-occurs with a strong anticyclone over Iceland and resembles the Greenland blocking weather regime[30], bringing cool and wet conditions over the Iberian Peninsula and over most of central and northern Europe (Fig. 1b, c). Conversely, the regions east of the JSL, i.e., southeastern Europe, experience hot and dry summers under this weather regime.

In contrast, the prevailing westerly flow over the North Atlantic is deflected northward (northwestern JSL) when an atmospheric ridge is centered over the British Isles and extends into northern and central Europe (European blocking weather regime; Fig. 1d). The ridge and the northwestern summer JSL anomaly promote warm and dry weather conditions over northern and central Europe, whereas summers in southern Europe, and particularly the Balkans, are anomalously wet and cool (Fig. 1e, f). When blocking is located over Scandinavia and a cyclone over southern Europe diverts the westerly flow southward (southeastern JSL, Fig. 1g), Italy and the Balkans experience anomalously cool and wet conditions (Fig. 1h, i). This Scandinavian blocking weather regime, on the other hand, creates warm and dry anomalies over northern Europe.

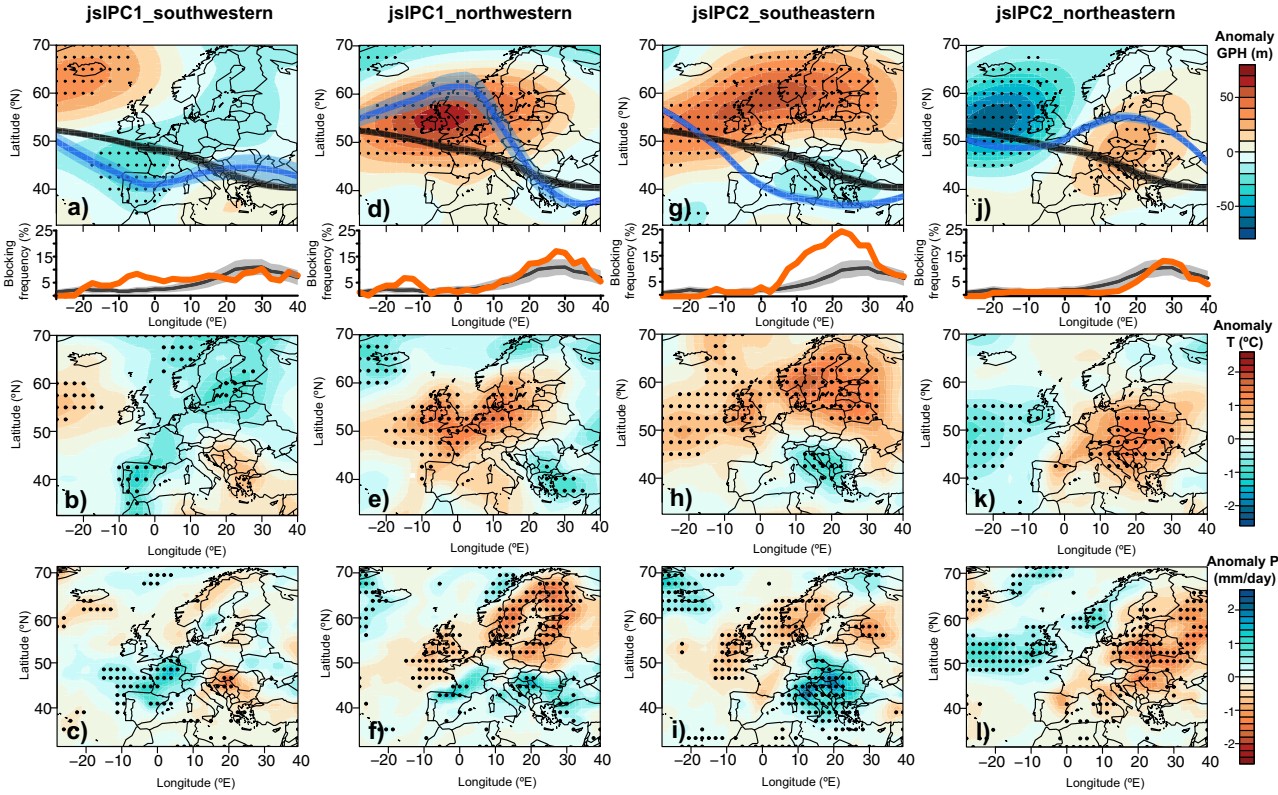

**Fig. 1 July–August composite climate anomalies during jet stream latitude (JSL) extreme years (i.e., D90 and D10 of jslPC1 and jslPC2 scores).** Maps represent averaged July–August anomalies during southwestern, northwestern, southeastern, and northeastern JSL extremes of 500 mbar geopotential height (GPH; m) (**a**, **d**, **g**, **j**), air temperature (T; °C) (**b**, **e**, **h**, **k**), and precipitation (P; mm/day) (**c**, **f**, **i**, **l**). Black dots represent significant ($p < 0.05$) departures from the long-term mean climatology. Line graphs in the top panels also show the mean July–August JSL position for the five extreme years (blue line) and standard error (blue shading) compared to the mean for the period 1950–2005 (black lines and shading). Orange lines represent the mean July–August blocking frequency per longitudinal section for the five extreme years compared to the mean for the period 1950–2005 (gray line). Gray-shaded areas around blocking frequency climatological mean correspond to two standard deviations from the mean.

Finally, northeastern JSL displacements occur when persistent anticyclonic conditions over Europe propel the JSL in a north-ward anomaly and a trough dominates the Atlantic east of Ireland (Fig. 1j). Summer weather conditions over the British Isles and western Scandinavia are anomalously cool and wet as they are influenced by the Atlantic trough weather regime, whereas warm and dry summers extend over most of the continent due to the anomalous anticyclonic circulation (Fig.1k, m).

**JSL impacts radial tree growth and forest productivity.** The JSL-driven summer weather dipole between northwestern and southeastern Europe is a recurring and prominent climatic feature that is reflected in the spatial pattern of European beech radial growth and simulated temperate forest GPP anomalies across the continent (Fig. 2).

Forest productivity, and radial tree growth in particular, in southeastern Europe, is most strongly influenced by southward migrations of the summer JSL (Fig. 1a, g; Fig. 2a, b, e, f), whereas productivity and radial tree growth in central and northern Europe is most strongly impacted by northward summer JSL anomalies (Fig. 1d, j; Fig. 2c, d, g, h). Specifically, southwestern summer JSL anomalies that co-occur with the Greenland blocking weather regime produce the largest radial tree growth and GPP reduction in southeastern Europe (up to 38% and 34%, respectively; Fig. 2a, b). These reductions offset the simultaneous non-significant increases (up to 16% for radial tree growth and 25% for GPP) in northwestern and Central Europe (Fig. 2a, b). The spatial pattern of tree radial growth and GPP anomalies is

reversed during southeastern summer JSL anomalies that are linked to a persistent continental-Scandinavian blocking weather regime (Fig. 1g–i). In southeastern Europe, European beech radial growth and GPP are boosted significantly during the associated cooler and wetter summers by up to 33% and 36%, respectively (Fig. 2e, f), whereas warmer and drier conditions in northwestern Europe led to a less conspicuous productivity decrease of up to 24% and 10% in radial tree growth and GPP.

During northwestern JSL, prevailing cyclonic conditions in southeastern Europe lead to cooler and wetter weather that results in European beech productivity and radial tree growth increases in this region of up to 25% and 22%, respectively (Fig. 2c, d). In Central and Northern Europe, forest productivity and specifically European beech radial growth decrease most strongly due to the warmer and drier weather originating from the atmospheric ridge (up to 38% and 17% significant reduction for radial tree growth and GPP). This decrease in forest productivity in central and northern Europe is counteracted by large increases during northeastern summer JSL extremes (Fig. 1k, m).

The cool and wet anomalies associated with the Atlantic trough weather regime during northeastern summer JSL anomalies benefit European beech growth and GPP in the British Isles and extend to central Europe (up to 29% and 20% significant increase, respectively; Fig. 2g, h), but result in radial tree growth and GPP reductions of up to 22% and 26% in the Mediterranean fringes of the study area, where warm and dry anomalies prevail.

These regional differences observed in European beech's climate sensitivity can be attributed to the stronger limiting effect of warm anomalies at the warm edge of the species

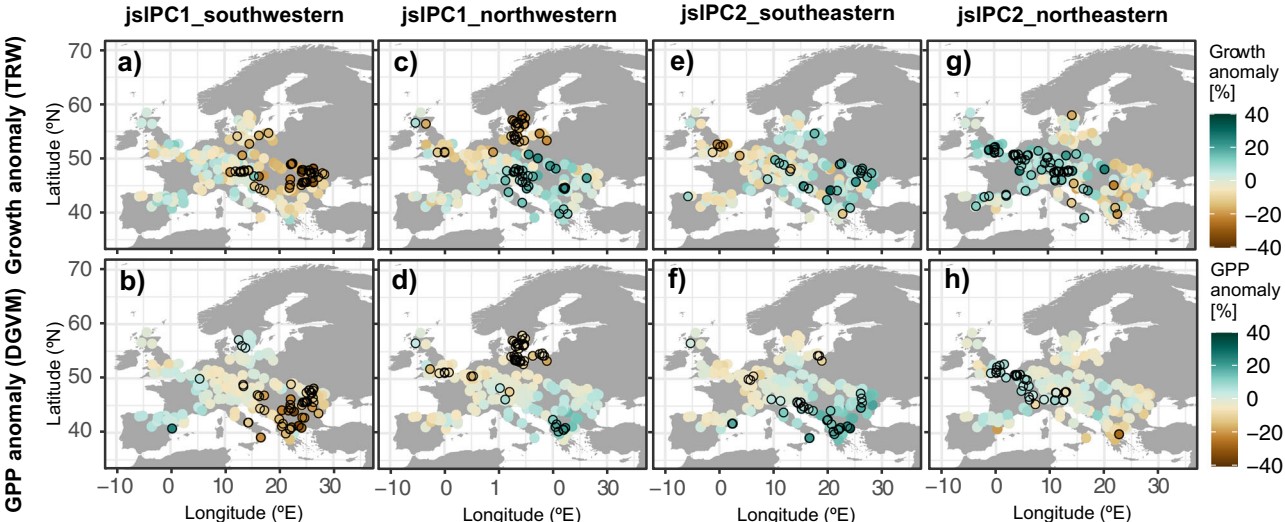

**Fig. 2 European beech radial growth and forest productivity during summer JSL extremes.** Anomalies in European beech radial growth (**a**, **c**, **e**, **g**) and DGVM-estimated forest GPP (**b**, **d**, **f**, **h**) for the years showing the largest summer JSL anomalies (i.e., D90 and D10 of jslPC1 and jslPC2 scores, see Supplementary Table 1). Anomalies are expressed as percentages from the mean for the period 1950–2005. Black circles indicate significant anomalies ($p < 0.05$).

distribution in the Mediterranean, compared to the core and cold edge of the species distribution in central and northern Europe (Supplementary Figure 3).

Radial tree growth and GPP mostly show spatially synchronized changes in response to JSL and weather regime variability, reflecting common climatic constraints to both processes[31,32]. Despite this synchronicity, radial tree growth displays a stronger sensitivity to extremes in climate than GPP. Among the potential reasons driving these differences in sensitivity is the fact that radial tree growth integrates complex multiyear signals through carbohydrate allocation and storage[33] and is inherently more sensitive to water availability than GPP[17,21]. Indeed, radial tree growth and photosynthesis may differ in environmental drivers and the first can even exert some control over photosynthesis through internal feedbacks[34].

The coupling between European beech radial growth and variations in the summer JSL is further demonstrated by correlation maps between radial tree growth across the European beech TRW network and the two main modes of summer JSL variability (Fig. 3). Both modes of summer JSL variability show links with radial tree growth across the continent, but with opposite signs in northwestern versus southeastern Europe (Fig. 3a, b). This is the most pronounced for the correlation between individual TRW site chronologies and the second mode of JSL variability (jslPC2; Fig. 3b). Interannual variability in this mode is negatively correlated with interannual variability in radial tree growth in northwestern Europe, but positively correlated with radial tree growth in southeastern Europe.

In fact, this JSL-driven radial tree growth dipole is so dominant that it is also a prominent feature of the two main modes of radial tree growth variability across Europe (Fig. 3c, d). The primary mode of European beech radial growth variability (trwPC1; 19% variance explained) reflects a dipole between northern, temperate Europe (positive loadings) and southern, Mediterranean Europe (negative loadings), with the strongest negative loadings found in Italy and the Balkans (Fig. 3c). The second mode (trwPC2; 10% variance explained; Fig. 3d) mirrors the dipole correlation pattern associated with the main mode of summer JSL variability (Fig. 3a) even more strongly and shows a continental dipole in European beech radial growth between northwestern and southeastern Europe, with a pivot point centered on the Alps (Fig. 3d). The

resemblance between the patterns in Fig. 3a and d versus Fig. 3b, c is further demonstrated by significant correlations between the corresponding time series (Supplementary Figure 4). The subsequent modes of radial tree growth explain the declining percentage of common variability (7% for trwPC3 and 5% for trwPC4).

Overall, the link between radial tree growth extremes and summer JSL extremes confirms the key role of the summer JSL as a driver of European beech radial growth across its distribution range. In particular, the two sets of extremes (JSL and radial tree growth) have almost half (nine out of nineteen) of their years in common (Supplementary Table 1), suggesting convergence between ecological and climate extremes.

The dominant role of summer JSL variability in driving contrasting regional European beech radial growth across Europe is confirmed and quantified by a linear mixed-effects model (LMM; see Methods Supplementary Figures 5–7). The radial tree growth at the 344 European beech forest sites that comprise our network is modeled as a function of the two main modes of summer JSL variability for the current and previous years, thus integrating legacy effects[35]. The selected LMM (Supplementary Equation 1), considering fixed and random effects (conditional $R^2$[36]), explains 32% of the total interannual radial tree growth variability (1950–2005) and up to 46% of extreme years. Generally, the LMM is more effective at simulating European beech radial growth during western extremes of summer JSL variability ($R^2 = 0.37$) compared to eastern extremes ($R^2 = 0.25$; Supplementary Figure 6). Its effectiveness is, however, even greater when simulating growth during extreme years of the main modes of radial tree growth variability (rather than JSL variability; Fig. 4; Supplementary Figure 7). The model explains 46% and 33% of variability during extremes of the primary (trwPC1) and secondary (trwPC2) modes of radial tree growth variability, respectively (Fig. 4b, e, h, k).

Similar to the composite of radial tree growth and GPP anomalies during summer JSL extremes (Fig. 2), composites for extremes of the main modes of radial tree growth variability (Fig. 3c, d) also show northwest-southeast polarity (Fig. 4). This polarity is pronounced across TRW, LMM-simulated radial tree growth, and GPP anomalies (Fig. 4). The main mode of European beech radial growth variability is expressed most strongly in

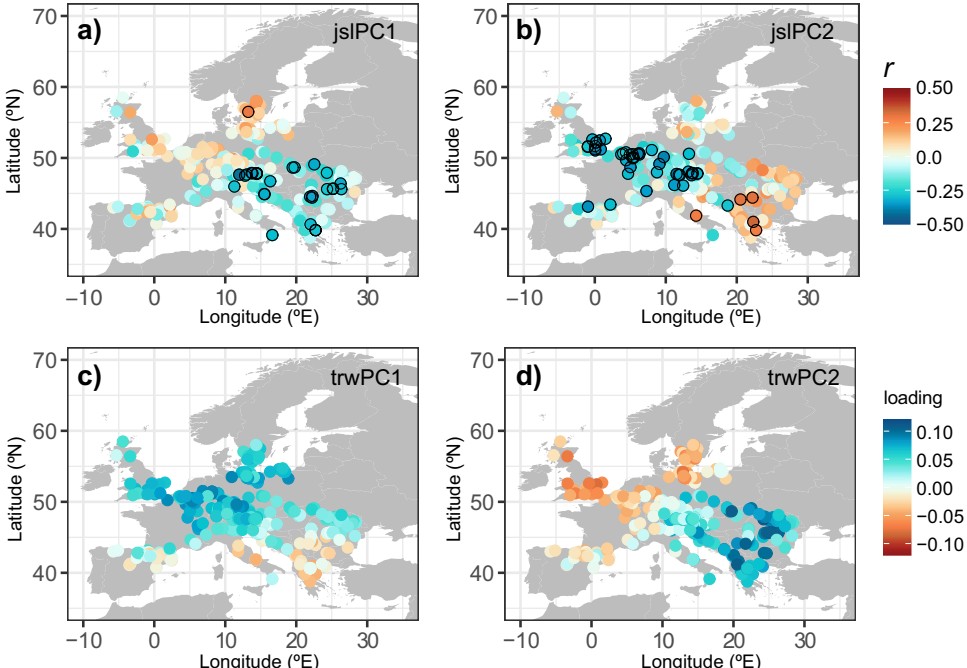

**Fig. 3 Spatial patterns of radial tree growth in relation to summer JSL variability.** Correlation map of European beech chronologies and the scores of jslPC1 (**a**) and jslPC2 (**b**). The bottom panels show the spatial pattern of loadings for the first (**c**) and second mode (**d**) of radial tree growth variability (trwPC1 and trwPC2). Black circles in the upper panel indicate significant correlations ($p < 0.05$). Analyses were performed for the period 1950–2005.

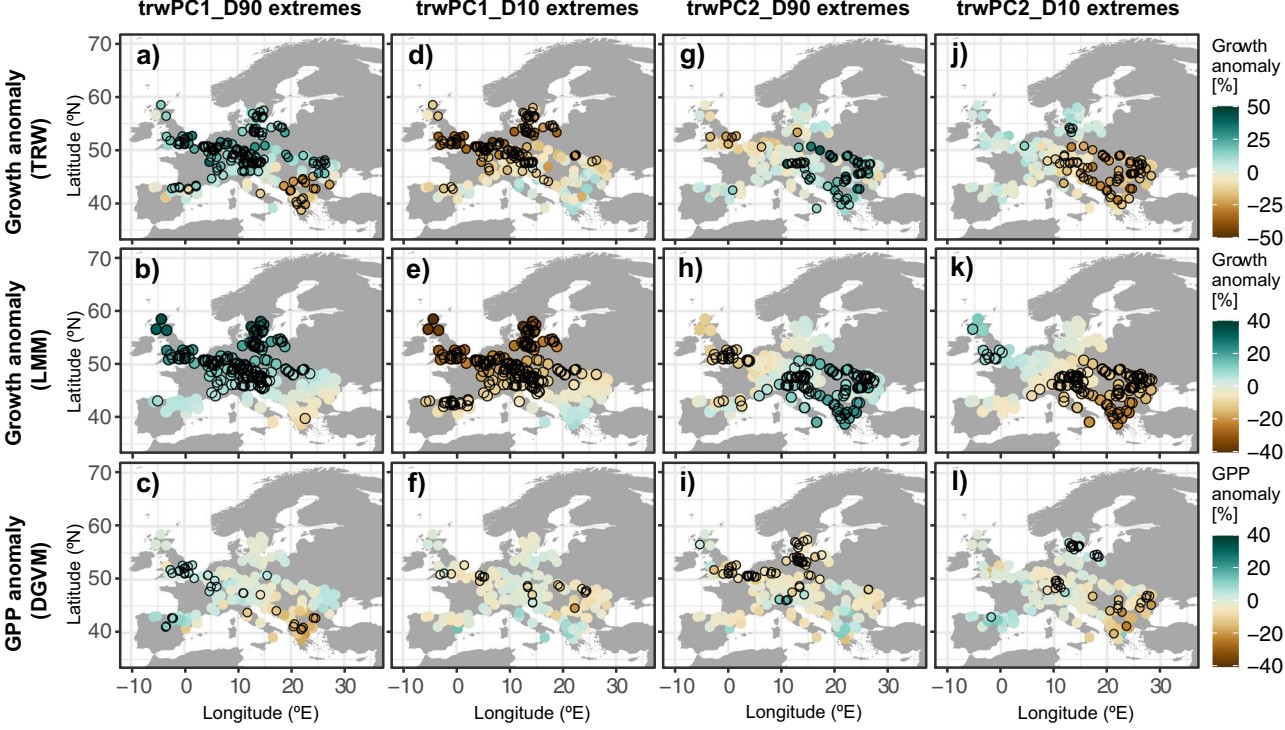

**Fig. 4 European beech radial growth and forest productivity during TRW extremes.** Anomalies in radial tree growth (TRW, **a**, **d**, **g**, **j**), simulated anomalies in radial tree growth by the LMM (**b**, **e**, **h**, **k**), and DGVM's simulated anomalies in forest GPP (**c**, **f**, **i**, **l**) for the years showing the largest radial tree growth anomalies (i.e., D10 and D90 of trwPC1 and trwPC2 scores, see Supplementary Table 1, Fig. 3). Anomalies are expressed as percentages from the mean for the period 1950–2005. Black circles indicate significant anomalies ($p < 0.05$).

northern and central Europe, with anomalies of up to 40% increase (50% decrease) in growth and 13% increase (20% decrease) in GPP (Fig. 4a–c, d–f). This main mode of radial tree growth is linked to the second mode of summer JSL variability (Supplementary Figures 8, 4), with years of particularly strong

radial tree growth increases in northern Europe linked to northeastern summer JSL positions (Fig. 4, and Supplementary Figure 8; see also Fig. 2). Unlike radial tree growth, GPP does not show any clear spatial pattern for the negative extremes of the main TRW mode (trwPC1_D10; Fig. 4f). The second mode of

radial tree growth variability is most evident as radial tree growth anomalies in southeastern Europe (Fig. 4g, h, j, k). Positive extremes of this second mode (trwPC2_D90) are related to northwestern summer JSL positions (Supplementary Figures 8g, 4a), resulting in positive radial tree growth departures over southeastern Europe (up to 43% increase; Fig. 4g). The GPP pattern, on the other hand, displays a larger number of significant negative anomalies at the other end of the dipole in north-central Europe (up to 16% decrease in GPP; Fig. 4i), suggesting differences in the impact of the same atmospheric configuration on forest carbon uptake versus radial tree growth. Negative extremes of this mode (trwPC2_D10) result in severe reductions over southeastern Europe in both radial tree growth and GPP of up to 33% and 22%. Despite the strong spatial dipole pattern, these are the only extremes that do not correspond to summer JSL deviations over the European domain (Supplementary Figure 8j–m; Fig. 4j–m). Overall, the LMM robustly reproduces the dipole pattern of radial tree growth across Europe during extremes. However, the model skill is better when simulating the magnitude (significance) of the positive radial tree growth anomalies than when simulating the negative ones (Fig. 4b, e, h, k, Supplementary Figure 6). This result is not unexpected, considering that the influence of the JSL modes on radial tree growth is most significant in one center of the dipole at a time, i.e., either in northwestern or southeastern Europe (Fig. 3a, b).

**The role of summer atmospheric blocking**. Northwestern, southeastern, and southwestern JSL displacements are to different extents linked to blocking regimes over the North Atlantic–European sector (Fig. 1) and thus unravel the relevance of these mechanisms for JSL and for persistent surface summer weather anomalies. In winter, the link between European/Scandinavian blocking and JSL has been controversial and related to both northward and southward JSL displacements[1,37]. We show that also in summer, European/ Scandinavian blocking can be associated with either northeastern (Supplementary Figure 8a) or southeastern JSL displacements (Fig. 1g) and that these displacements have opposite impacts on forest productivity. It is therefore the displacement of the eastern JSL north or south of the blocking center that will determine the sign of the radial tree growth polarity across Europe. Indeed, European/Scandinavian blocking coupled with a southeastern JSL produces the largest productivity increase in southeastern Europe (33% and 36% in radial tree growth and GPP, respectively; Fig. 2e, f). Conversely, European/Scandinavian blocking coupled to a northeastern JSL is related to significant increases in radial tree growth up to 40% in central-northern Europe (Fig. 4a; Supplementary Figure 8a). The physical link between summer JSL and European/Scandinavian blocking is particularly relevant considering that up to 80% of summer heatwaves in northern Europe are associated with blocking events[38] and that European/Scandinavian blocking is more frequent than Greenland blocking, with the highest percentage of blocked days in summer occurring between 15–35°E[8,39].

## Discussion
Changes in ecosystem productivity in the North Atlantic–European domain have been linked to the strength and sign of the main modes of atmospheric variability (i.e., North Atlantic Oscillation (NAO) and the East Atlantic (EA) pattern) as a way to synthesize ecosystem responses to climate variability[40,41]. Indeed, we show that the first and second modes of radial tree growth variability in our European beech network are linked to these dominant teleconnection patterns in summer over Europe that is also related to

the dominant modes of JSL variability (Supplementary Figure 9). Previous studies showed that jet stream movements, however, integrate spatiotemporal variability not accounted for by the main variability modes of the atmospheric circulation alone[1,29]. We find that a summer JSL framework provides a tangible and physically measurable representation of the atmospheric state that can be connected to changes in forest productivity and growth.

Our study emphasizes the key role that summer JSL variability plays in driving the European beech dipole of carbon uptake (GPP) and accumulation (radial tree growth), depending on the longitudinal window of JSL displacements and the co-occurrence of summer weather regimes with the location of cyclonic and anticyclonic systems or blocking centers[8,42]. Summer climate is a ubiquitous driver of radial tree growth variability in European beech across its distribution range[43,44]. The influence of JSL on summer surface weather and radial tree growth represents one-third of the radial tree growth variability across the network and is responsible for up to 50% of radial tree growth change during extreme years. The strength of the effect of summer JSL variability on European beech radial growth varies spatially, with typically stronger impacts at the species distribution edges. In these regions, European beech populations are closer to their limits of tolerance to climate and respond adaptively to avoid surpassing their physiological thresholds during summer heatwaves or drought events[45]. Other climatical and ecological factors may explain a variable percentage of radial tree growth variability in our network (e.g., Supplementary Figure 10). Under anthropogenic climate change, spring temperature is increasingly more relevant for populations at the core and cold edge of the European beech distribution range, whereas warm-edge forests are more sensitive to winter temperature[46,47]. While we show that summer JSL had the strongest effect during the second half of the 20th century, future JSL changes in other seasons than summer may therefore become increasingly relevant at regional scales, particularly where trees are most sensitive to forthcoming changes in climate.

Forecasting the evolution of the physical coupling between JSL variability, the alteration of surface weather, and extremes in terrestrial productivity for the 21st century is of great scientific and socio-economic interest. Accurate projections of the frequency of large-scale summer heatwaves and droughts rely on our capacity to simulate key dynamical components of the general atmospheric circulation, including jet stream position and strength, storm tracks, and atmospheric blocking[6,7,48]. Some studies have suggested a recent increase in jet stream waviness and in the related frequency of blocking events[49,50]. Future projections of jet stream dynamics, however, are highly uncertain[51,52] due to biases in the model representation of jet stream flow variability, and particularly changes in the transition from a zonal to a blocked flow[37,53,54] and the debated role of Arctic Amplification in driving mid-latitude circulation and extremes[2,52,55]. The blocking frequency over the Atlantic and Scandinavia is expected to remain constant or decrease under increasing greenhouse gas concentrations[8,53], but this projected decrease does not imply a shorter extreme event duration. On the contrary, the same projections point to less frequent but more persistent blocking events[53]. Furthermore, Eurasian blocking events (i.e., 50E and further east) are projected to increase and should be taken into account since they may affect the easternmost part of Europe[56], as was the case during the 2010 Eurasia summer heatwave[7,39]. Still, years of documented extreme summer heatwaves in Europe such as 2003 ([28]for a complete list of extremes) were detected neither as a year of extreme JSL, nor as a year of extreme radial tree growth across Europe. Atmospheric configurations giving rise to summer heatwaves such as the so-called omega blocking[57] usually involve a higher number of

pressure centers and a wavier jet stream (i.e., a higher number of meanders or waves)[50] that may not be always detected by our JSL approach. Further, most high-impact extreme events are not the sole result of the influence of a single driver such as the variability of simultaneous large-scale atmospheric circulation, but they occur due to a combination of different processes that interact[12,13,22]. Anomalous large-scale circulation, oceanic or soil-moisture conditions during the previous seasons, or other processes at lower atmospheric levels might also play a crucial role in the occurrence of these events.

Future differences in the European beech forest productivity and radial tree growth dipole will further be modulated by the portion of the variability not explained by summer JSL. Among the factors integrating that variability are regional differences in the trees' climate sensitivity to other seasons[44,46], legacy effects[58], site conditions that buffer against climate variabilities such as forest composition[59], genetic composition due to past demographic and phylogeographic processes[60], and historical responses to forest management and disturbances[61]. Southeastern European beech forests are closer to their physiological limits (i.e., the warm/dry edge of their continental distribution), but they show a more plastic response to climate compared to forests at the core and cold edge of the distribution range (central-northern Europe)[44,62]. Core range populations generally grow under near-optimal climate conditions, but still respond distinctly to the occurrence of dry spells[15] due to a low resistance to summer drought that is compensated by a high resilience[44]. The impact of summer weather extremes on the European beech forests growing at the warm edge is more ambiguous[44,62], but the resistance-resilience trade-off is a range-wide essential component of the trees' capacity to buffer against long-term climate change. Climate-related mortality risk in angiosperms is more likely when trees show low resistance to previous drought episodes[63]. Thus, the lower resistance of the population at the core range to summer weather extremes may imply higher climate-related mortality risk under frequent or persistent European/Scandinavian blocking weather regimes (i.e., northeastern and southeastern JSL movements).

In addition to this, legacy or lagged climatic effects on radial tree growth are common in deciduous species and might become increasingly significant in the future because they are enhanced when climatic conditions become strongly limiting[64]. Temperature or drought-induced stomatal closure in summer reduces photosynthesis and lowers the non-structural carbon pool to support growth onset and leaf formation early in the following growing season[65]. Furthermore, limiting conditions during the previous summer cue massive seed production in European beech (i.e., masting) in the following growing season, and such a reproductive effort may reduce the carbon reserves available for growth[64,66]. Despite being recognized as an essential component to understanding the impacts of climate change on terrestrial ecosystem productivity and carbon cycling, growth-reproduction trade-offs are still not implemented or sufficiently reproduced by DGVMs[34,35,58], which may explain the differences we find between anomalies of radial tree growth and GPP.

Our results showcase that the impacts of JSL displacements on European beech forest productivity are not uniform across the continent. In particular, the net effect of the JSL-driven summer climate dipole on European beech radial growth and carbon uptake dramatically differs between the eastern versus western longitudinal windows, where modes of JSL displacement primarily occur within the North Atlantic–European domain. The productivity imbalance generated by the JSL displacements entails important implications for risk trade-offs in forest intervention and management planning aimed at forest preservation or mitigation of carbon emissions[67]. Counteracting the continental

imbalance of forest productivity in Europe will largely depend on forest structure, density, natural and anthropogenic-induced disturbance dynamics, and physiological adaptations at the species and population levels. The net effect of JSL variability on continental-scale terrestrial carbon fluxes may therefore co-depend on increasing forest resilience, as well as on balancing carbon stocks and rates of forest productivity.

## Methods

**Climate data.** We calculated the monthly mean JSL over the North Atlantic–European domain (30W–40E) using the NCEP/NCAR reanalysis 1 product (1948–2018; 2.5° × 2.5°)[26]. We selected the NCEP/NCAR reanalysis product because it covers the target period of analyses 1950–2005 and has been extensively used in the characterization of North Atlantic jet stream variability (e.g., Davini and D'Andrea[68]). We defined monthly JSL for each 2.5° longitudinal window as the latitude (20–90 N) at which the monthly averaged 300 mbar zonal wind speed is at its maximum[69]. We focused our analysis on monthly JSL averaged for July and August (summer). In order to link summer JSL extremes to climatic patterns over the North Atlantic–European domain, we also retrieved monthly and seasonal (July and August) air temperature, precipitation, and 500 mbar geopotential height (GPH) fields from the NCEP/NCAR reanalysis product. To calculate the frequency of blocking events, we analyzed daily NCEP/NCAR 500 mbar GPH fields and applied the blocking detection method developed by Doblas-Reyes et al.[70]. Blocking frequency was defined as the percentage of blocked days at a given longitude relative to the total number of days for the summer (July and August) season.

**Tree-ring data.** European beech (*Fagus sylvatica* L.) is one of the dominant temperate tree species in Europe and is highly sensitive to drought across its distribution range[44]. To analyze the link between European beech radial growth and JSL at the continental scale, we made use of the European Beech Tree-ring Network (EBTRN). The EBTRN is a unique database of TRW measurements derived from beech forests that contains more than 600 chronologies. We defined "forest" broadly in this study and include all forested areas dominated by European beech trees. The definition covers any land dominated by trees and includes European beech as the main species or one of the dominant tree species. The network covers the full range of beech distribution in Europe and spans a maximum temporal domain from 1468 to 2018. To analyze growth extremes at the continental scale, we used a subset of the EBTRN network that was balanced in terms of spatial and temporal domain coverage. The subset consists of 9402 individual TRW series grouped from 344 site chronologies, covers the distribution range of European beech, and spans a common period 1950–2005 (Supplementary Figure 1). Each TRW chronology in the subset represents European beech radial growth at a given forest (site) and is composed of individual TRW measurements of 21 trees on average. The network includes both pure and mixed forests with different management histories and disturbance dynamics. We, therefore, applied a flexible detrending method that responds to the need to remove age-related growth trends and minimize non-climatic influences[71]. We detrended the individual TRW series using a 32-years cubic-smoothing spline and developed a chronology for each site using a bi-weight robust mean. Trees across the EBTRN network are most sensitive to climate conditions during the summer[43] (see also Supplementary Figure 10) and we thus selected this season for analysis.

**DGVM simulations of GPP.** To quantify carbon uptake by the forest ecosystems at our study sites, we obtained monthly GPP simulated by the TRENDY model ensemble Version 6[72] for the period 1901–2016. The spatial resolution of the models ranges from 0.5° to 2°. We chose the GPP output of only the six DGVMs containing a "temperate broadleaf deciduous" plant functional type (PFT) over the overall GPP, because PFTs vary strongly between models and not all models have a PFT that corresponds to that of European beech. For each model, we extracted summer (July–August) GPP for the "temperate broadleaf deciduous" PFT from the grid cells that the site coordinates fall into, acknowledging that geographically close sites may fall into the same grid cell. Next, we calculated percent anomalies in summer GPP for each site and model as the deviation of summer GPP from average. We approximated the average GPP for a given time period by a cubic spline function with a 50% frequency cutoff at 10 years that was fitted to the GPP time series. We selected this flexible fit to capture short-term anomalies in extreme years and remove longer-term trends in GPP. Finally, we averaged the percent anomalies in GPP from all models for the positive and negative extreme years that emerged from the analysis of JSL (Supplementary Table 1).

**JSL, climate, and forest growth analyses.** To determine the main modes of summer JSL variability over the North Atlantic–European domain, we applied a principal component analysis (PCA) to the time series (1950–2005) of July–August JSL for each 2.5° longitudinal band of the domain (Supplementary Figure 2). The first and second modes (i.e., jslPC1 and jslPC2) combined explained more than 55% of July–August JSL variability. We defined JSL extremes as the 1st (D10) and

9th (D90) deciles of jslPC1 and jslPC2 (Supplementary Table 1, Supplementary Figure 2).

To investigate the climate patterns typical for the two main JSL modes (jslPC1 and jslPC2), we composited July–August climate (air temperature, precipitation, and 500 mbar GPH) field anomalies, as well as summer frequency of blocking events for jslPC1 and jslPC2 extremes (Fig. 1). Field anomalies were computed for the North Atlantic–European domain as linearly detrended deviations from the seasonal climatology of the common period. The significance of field anomalies (p-value estimates) was computed by Monte–Carlo tests with 1000 permutations.

To analyze the influence of summer JSL on radial tree growth, we calculated Pearson's correlation coefficients between the jslPC1 and jslPC2 time series and each tree-ring chronology selected from the EBTRN (Fig. 3). To investigate tree growth and GPP anomalies during extreme JSL years, we composited the set of 344 TRW and GPP chronologies for the jslPC1 and jslPC2 extremes (Fig. 2). In order to provide a comparable output to the GPP anomalies retrieved from the DGVMs, tree-growth anomalies were converted into a percentage of radial tree-growth change with respect to the common period of analysis 1950–2005 prior to compositing. Composite maps represent the mean percentage of European beech radial growth (TRW) and productivity (GPP) change during extreme years per site. The significance of anomalies (p-value estimates) was computed by Monte–Carlo tests with 1000 permutations.

We also applied a second PCA to the set of 344 TRW chronologies to determine common growth variability among sites. The first two PC modes of TRW (i.e., trwPC1 and trwPC2) combined explained almost 30% of European beech growth variability. In line with the JSL analysis, we defined tree-growth extremes as the 1st (D10) and 9th (D90) deciles of the trwPC1 and trwPC2 time series (1950–2005; Supplementary Table 1). We then composited GPP and modeled radial tree growth for trwPC1 and trwPC2 (D10 and D90) extremes (Fig. 4). Similar to Fig. 2, composite maps represent the mean percentage of radial tree growth (TRW) and productivity (GPP) change during tree-growth extremes per site, and the significance of anomalies was computed by Monte–Carlo tests with 1000 permutations.

We further explored the connection between radial tree growth and JSL using LMMs. We developed LMMs to explain radial tree growth of the 344 tree-ring site chronologies for the period 1950–2005, including various combinations of fixed and random factors. As fixed effects, we considered the main modes of summer JSL (jslPC1 and jslPC2), as well as continuous variables such as longitude, latitude, and elevation (Supplementary Tables 2, 3). The scores of jslPC1 and jslPC2 lagged by one year (i.e., $jslPC1_{y-1}$ and $jslPC2_{y-1}$) were also included as predictors to account for the legacy effects of the previous year's climate on the current year's tree performance[41,66]. All predictors were standardized and we selected models based on an ANOVA test including AICc (corrected Akaike Information Criterion), with lower AICc values indicating a stronger explanatory power.

We first selected the variables with a significant effect on radial tree growth across the network of chronologies (i.e., explanatory variables). From the seven fixed effects considered in this study (Supplementary Table 2), predictors displaying a p value >0.05 were excluded from the model (Supplementary Table 3). The relevance of the excluded and remaining fixed effects are confirmed by the ΔAICc (Supplementary Tables 2, 3), defined here as the difference of AICc between the full model (i.e., including all seven fixed effects) and the model without the predictor of interest. Collinearity among predictors was checked with the variance inflation factor (VIF). We used a conservative threshold and predictors displaying VIF values >2 were discarded from the model.

The effect of JSL on radial tree growth may differ across locations and years (Figs. 2, 4). In order to account for these different effects and the non-independence among measurements, we tested both variables (site and year) as grouping factors (random intercepts) and different combinations of the seven predictors tested as fixed effects, as random slopes. Starting from a saturated model with the three fixed-effect or explanatory variables selected in the previous step, we created a fully crossed set of models, including different combinations of random slopes for each grouping factor and considering interactions among variables (Supplementary Table 4). Random slopes were progressively discarded from the model according to the model AICc until AICc did not further decrease despite lower model complexity (i.e., when models with a lower number of degrees of freedom displayed reductions in AICc <2). Interactions among predictors and random slopes did not improve model results. Collinearity among predictors was checked with VIF to ensure that variables displayed a VIF <2 in the final model selection (Supplementary Table 4; Supplementary Equation 1). We evaluated the predictive skill of the selected model using the leave-one-out cross-validation scheme. Model quality was checked by inspecting residual patterns and looking for signals of heteroscedasticity, non-normal distribution, and autocorrelation (Supplementary Figure 5).

All analyses were conducted using Matlab R2017b and R software version 4.0.2[73] and R packages dplR[74], lme4[75], lmerTest[76], and MuMIn[77].

## Data availability

The European beech GPP simulations by the TRENDY model ensemble Version 6 and the subset of chronologies from the EBTRN data used in this study are available in the Figshare database under accession code [https://doi.org/10.6084/m9.figshare.c.5660008].

The climate data sets for JSL calculation and GPH, air temperature and precipitation composite maps are available in the NCEP-NCAR website repository (https://www.esrl.noaa.gov/psd/data/gridded/data.ncep.reanalysis.html).

## Code availability

We used R software for computations and visualization. The R libraries used are specified in Material and Methods. Custom R-script code for JSL calculation is available at https://doi.org/10.6084/m9.figshare.c.5660008.

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

## Acknowledgements

This work was supported by Fundació La Caixa through the Junior Leader Program (LCF/BQ/LR18/11640004) and the Universidad Politécnica de Madrid through the Programa Propio (PINV-18-SBSYN2-105-F1TXYR). The following authors acknowledge funding support. I.D.L.: Agnese N. Haury Visiting Scholar & Trainee Fellowship (Laboratory of Tree-Ring Research, University of Arizona), the Mobility Award José Castillejo, Ministry of Education, Spanish Government (CAS19/00331) and the Programa de Ayudas Beatriz Galindo, Secretaria de Estado de Universidades, Investigación, Desarrollo e Innovación (#BG20/00065). V.T.: National Science Foundation CAREER grant (AGS-1349942). B.A.: Spanish Ministry of Science and Innovation through the JeDiS project (RTI-2018-096402-B-I00). F.B.: project "Inside out" (#POIR.04.04.00-00-5F85/18-00) funded by the HOMING program of the Foundation for Polish Science co-financed by the European Union under the European Regional Development Fund. AB, AM, CSZ: Bavarian Ministry of Science and the Arts in the context of the Bavarian Climate Research Network (BayKliF). A.H.: PinCaR project (UHU-1266324) by ERD Funds, Andalusia Regional Government, Consejería de Economía, Conocimiento, Empresas y Universidad 2014-2020. EM-S: Swiss National Science Foundation project

TRoxy (No. 200021_175888). A.S.J.: Natural Environment Research Council grants NE/V00929X/1 and NE/S010041/1. J.K., L.M., M.M.T., R.W., M.W.: research training group RESPONSE funded by the German Research Council (DFG Fi 846/8-1, DFG GRK2010). AMP: Romanian Ministry of Research, Innovation, and Digitization, Project-PN-19070506/Ctr. no. 12 N/2019. I.C.P.: grant of the Romanian Ministry of Education and Research, CNCS-UEFISCDI within PNCDI III (PN-III-P4-ID-PCE-2020-2696). R.S.S.: DendrOlavide I (EQC2018-005303-P), Ministry of Science, Innovation and Universities, Spain; DendrOlavide II (IE19_074 UPO), VURECLIM (P20_00813) and VULBOS (UPO-1263216). T.L.: Slovenian Research Agency—research core funding no. P4-0107 Program research group "Forest Biology, Ecology and Technology". We thank Virgilio Gómez-Rubio for assistance and advice on the LMM development. We thank Christoph Dittmar, Wolfram Elling, and numerous students of the University of Applied Sciences Weihenstephan-Triesdorf for providing European beech tree-ring chronologies.

## Author contributions
I.D.L. and V.T. conceived the idea. I.D.L., B.A., and F.B. performed the analysis with the participation of G.X. and V.T. I.D.L. and V.T. led the manuscript writing with significant contributions from B.A., F.B., and G.X. All authors provided data and commented on different versions of the manuscript. C.S.Z. manages and updates the EBTRN database.

## Competing interests
The authors declare no competing interests.

## Additional information

[1]Dpto. de Sistemas y Recursos Naturales, Universidad Politécnica de Madrid, Madrid, Spain. [2]Dpto. Física de la Tierra y Astrofísica, Universidad Complutense de Madrid, Madrid, Spain. [3]School of Natural Resources and the Environment, University of Arizona, Tucson, AZ 85719, USA. [4]Laboratory of Tree-Ring Research, University of Arizona, Tucson, AZ 85721, USA. [5]State Key Laboratory of Cryospheric Sciences, Northwest Institute of Eco-Environment and Resources, Chinese Academy of Sciences, Lanzhou 730000, China. [6]Department of Environmental, Biological and Pharmaceutical Sciences and Technologies, University of Campania Luigi Vanvitelli, 81100 Caserta, Italy. [7]Land Surface-Atmosphere Interactions, Technical University of Munich, Freising, Germany. [8]Faculty of Forestry and Wood Sciences, Department of Forest Ecology, Czech University of Life Sciences, Prague, Czech Republic. [9]Pyrenean Institute of Ecology, (IPE-CSIC), Zaragoza 50059, Spain. [10]Biological and Environmental Sciences, University of Stirling, Stirling, Scotland FK9 4LA, UK. [11]Forest is life, Gembloux Agro-Bio Tech, University of Liege, Gembloux, Belgium. [12]Forest Research Institute & Southern Swedish Forest Research Centre (SLU), Lomma, Sweden. [13]Environmental Meteorology, University of Freiburg, Freiburg, Germany. [14]Institute of Wood Technology and Renewable Materials, University of Natural Resources and Life Sciences, Vienna, Austria. [15]Department of Geography and Planning, School of Environmental Sciences, University of Liverpool, Liverpool L69 7ZT, United Kingdom. [16]Nature Rings – Environmental Research and Education, Mainz, Germany. [17]Department of Agroforestry Sciences, University of Huelva, Campus La Rábida, Palos de la Frontera, 21819 Huelva, Spain. [18]Faculty of Forestry, University of Belgrade, Belgrade, Serbia. [19]Faculty of Forestry, University of Agriculture in Krakow, Krakow, Poland. [20]Institute of Botany and Landscape Ecology, Greifswald University, Greifswald, Germany. [21]Institute of Biology, University of Hohenheim, Stuttgart, Germany. [22]University of Applied Forest Sciences, Schadenweilerhof, Rottenburg am Neckar, Germany. [23]Department of Forest Yield and Silviculture, Slovenian Forestry Institute, Ljubljana, Slovenia. [24]University of Primorska, Faculty of Mathematics, Natural Sciences and Information Technologies, Koper, Slovenia. [25]Chair of Forest Growth and Woody Biomass Production, TU Dresden, Dresden, Germany. [26]Swiss Federal Institute for Forest, Snow and Landscape Research WSL, 8903 Birmensdorf, Switzerland. [27]TUM School of Life Sciences, Ecoclimatology, Technical University of Munich, Freising, Germany. [28]Institute for Advanced Study, Technical University of Munich, 85748 Garching, Germany. [29]Dep. Agricultural, Forest and Food Sciences (DISAFA), University of Turin, Turin, Italy. [30]Plant Ecology, Albrecht-von-Haller-Institute for Plant Sciences, Georg-August-University Goettingen, Goettingen 37077, Germany. [31]Department of Earth and Environmental Sciences, University of Pavia, Pavia, Italy. [32]University of Forestry, Sofia, Bulgaria. [33]National Institute for Research and Development in Forestry "Marin Drăcea", Voluntari, Romania. [34]Transilvania University of Brasov, Brasov, Romania. [35]Center for Mountain Economy - CE-MONT, Vatra Dornei, Romania. [36]Forest Biometrics Laboratory, Faculty of Forestry, "Stefan cel Mare" University of Suceava, Suceava, Romania. [37]Dpto. Sistemas Físicos, Químicos y Naturales, Universidad Pablo de Olavide, Sevilla, Spain. [38]Forest Growth and Dendroecology, University of Freiburg, Freiburg, Germany. [39]Faculty of Letters, University of Bucharest, Bucharest, Romania. [40]Faculty of Forestry Sciences, Agricultural University of Tirana, 1029Kodër-Kamëz, Tirana, Albania. [41]Department of Forestry, University of Applied Sciences Weihenstephan-Triesdorf, Freising, Germany. [42]Institute of Biodiversity and Ecosystem Research, Bulgarian Academy of Sciences, Sofia, Bulgaria. ✉email: isabel.dorado@upm.es

