## [Peer Review File · Nature Communications]

Reviewers' Comments:

Reviewer #1:

Remarks to the Author:

In their study, Dorado-Liñán et al. analyse tree ring width (TRW) chronologies to understand how weather variability associated with the Jet Stream location (JSL) affect vegetation growth. They then combine the analysis with vegetation models to derive impacts on primary productivity due to JSL variability. The manuscript is well written and well referenced, and the topic is quite relevant, given the uncertainties associated with summer extreme events and their impacts on vegetation [1]. However, I find that the dominance and causal links of JSL variability in controlling TRW and GPP are over-emphasised and not fully supported by the results in the study as is, especially since uncertainties are treated rather superficially. I explain my concerns and make specific suggestions below.

The modelling approach needs to be clarified:

The LMM model explains 46% and 33% of variability during extremes of the primary (trwPC1) and secondary (trwPC2); however, these two PCs explain, respectively 19% and 10% of TRW variance. This means that the LMM model would explain, at best, less than 10% of the variance in TRW. Qualitatively the model seems to be able to reproduce the spatial variability of the different JSL modes, which can be because of the use of geographical fixed effects (lat, lon, elev). In fact, if I read Table S4 well, the LMM with latitude/longitude and elevation as fixed effects is the one with lowest AICc, as highlighted in bold by the authors. Was this the model used to derive Figure 4? This is not clear at all to the reader. Nevertheless, the ANOVA analysis in Table S4 seems to invalidate the usefulness of JSL in explaining variations in TRW.

Finally, what were the "random effects" considered? Did the authors consider "site" as a random effect? This would be reasonable as there may be a high fraction of variance explained by site-specific factors that cannot be accounted in the model.

The authors focus on summer jet stream variability and concurrent impacts on TRW. They also include summer JSL from the previous year in their modelling to account for legacy effects. It is unclear why the authors did not include winter or spring JSL variability, since most summer droughts have been shown to have an important spring rainfall deficit component [2].

The comparison with GPP from DGVMs is also problematic, as GPP corresponds to assimilation, but does not necessarily translate to growth. A better metric for comparison with TRW would be NPP or changes in aboveground C veg, which should be available from model outputs. Additionally, the authors chose the overall GPP simulated by models, while they could have selected the PFT-specific output, which would be more sensible to compare with TRW from a single species. The authors explain that the choice was motivated by the fact that not all models include a PFT that can be related to European beech, but most models will include a temperate deciduous PFT. Refining the analysis by using PFT-specific GPP (or rather NPP as suggested above) might improve the comparison with results for TRW in Figure 4.

Finally, in the abstract, the authors state that JSL "provides a synthetic and robust physical framework that integrates climate variability not accounted by atmospheric circulation patterns". After this, one would expect a comparison with traditional atmospheric circulation indices, which have also shown strong relationships with weather variability and vegetation productivity in Europe, such as the NAO, East-Atlantic or Scandinavian Patterns [3], [4], including their winter phases.

Figure 2 and corresponding discussion: I wonder how significant are differences between composites in TRW and GPP both per site (analogue to the stippling in Figure 1) and in space. Specifically, jsIPC1_NW and jsIPC2_SE look spatially very similar, which might be explained by the fact that Temp and Precip anomalies for these two composites are roughly similar.

The correlation maps between TRW and Temp/Precip in Figure S2 are much more heterogeneous than the patterns in Figure 2. Is it possible that other variables, for example radiation and cloudiness [5] could be a predominant factor in higher latitude / energy limited regions? This could also explain with correlations in most sites are rather low (a correlation of 0.28, even if significant, is quite weak, it corresponds to less than 10% of variance explained!).

Figure 3: it is not clear how one can compare panels (a-b) with (c-d) without knowing how the trwPCs relate to the jsIPCs.

Small comment:

Figure S1 – there is no green, rather cyan.

[1] M. Reichstein et al., "Climate extremes and the carbon cycle," *Nature*, vol. 500, no. 7462, pp. 287–295, Aug. 2013, [Online]. Available: <http://dx.doi.org/10.1038/nature12350>

[2] E. M. Fischer, S. I. Seneviratne, P. L. Vidale, D. Lüthi, and C. Schär, "Soil moisture–atmosphere interactions during the 2003 European summer heat wave," *J. Clim.*, vol. 20, no. 20, pp. 5081–5099, 2007.

[3] L. Comas-Bru and F. McDermott, "Impacts of the EA and SCA patterns on the European twentieth century NAO-winter climate relationship," *Q. J. R. Meteorol. Soc.*, 2013, doi: 10.1002/qj.2158.

[4] A. Bastos et al., "European land CO2 sink influenced by NAO and East-Atlantic Pattern coupling," *Nat. Commun.*, vol. 7, 2016.

[5] X.-P. Song et al., "Global land change from 1982 to 2016," *Nature*, vol. 560, pp. 639–643, 2018, doi: 10.1038/s41586-018-0411-9.

Reviewer #2:

Remarks to the Author:

The paper submitted by Dorado-Liñan et al. assesses a relationship between jet stream position variability and beech growth using dendrochronology and forest productivity using a vegetation modelling approach. This is a novel and interesting test, with potentially widespread consequences for beech survival and beech forest functioning. I very much liked reading this innovative paper and think it is suitable for the readership of *Nature Communications*.

These are my comments (major and minor comments are mixed together, sorry for that):

*In the abstract: perhaps you can add a final concluding sentence focused on the consequences of your findings and/or implications for forest functioning?

*L128: I would already here briefly describe the methods and data sources of the climatological part of your analyses (if you decide to keep it in the main text, but see further).

*L132, L600 and elsewhere: does the word chronologies reflect to one tree ring series of one individual beech tree? I think not, but this remains unclear as that information is also not available in the Methods section on L.601 where you mention that your database comprises 344 chronologies and 9402 individual tree-ring width series? How many trees were then included in the study in the end? Confusingly, on L650 you then use the term '344 TRW chronologies'. Is each circle in Fig. 2 and 3 one tree? Could you add maps/histograms/graphs of the sampled trees, the length of the timeseries available, and the year of sampling?

*In the discussion, no reflection is made on the potentially moderating effects of tree mixtures on the impacts of extreme events. Indeed, a growing body of literature shows that drought/heat

impacts on (beech but also other species) forests are lower in multispecies forests that have a higher tree species richness (e.g. Sousa-Silva et al. 2018 Global Change Biology). Do you have information on the forest composition of the sampled beech forests across Europe and on co-occurring tree species? If that information is available, you could test the influence of that as a covariate in your models.

*I personally find the climatological part of the paper the least innovative and relatively descriptive. For instance, on L140 till L200 and in Fig. 1, it remains unclear to me whether the maps and descriptions represent single-year (or weekly/daily) examples of each typical position of the JSL? If that is the case, could you mention the date/year in which these situations occurred in Fig. 1? Without devaluing the scientific value of this climatological part, it very much reads like a 'textbook' text on jet stream variability. To me, the central and most innovative part is the relationship with the tree growth and forest productivity starting on L203. I understand you need the climatological introduction, but perhaps you can consider to move a large part, or even the majority of this section, to the Supplementary Information.

*In many of the figures, there are a lot of abbreviations used that are not explained in the captions. To improve readability of the captions, I would at least always mention the whole term as well.

*Code availability (L437): is there a reason not to upload the code together with the data in an online repository?

*The definition of summer does not include June (L580), in contrast to the meteorological definition of the northern hemispheric summer (JJA). Why did you also not consider June in your analyses? Was there a specific reason for this? In recent years, extreme droughts and heatwaves also occurred in June in Europe.

*L599: can you explain this subsetting procedure of the EBTRN network "in terms of spatial and temporal domain" a bit more? For instance, why does this subsetting procedure decrease your time window from 1468-2018 to 1950-2005? Is it because you omitted chronologies < 15 trees? Is this then < 15 trees within one site? Of what area?

Congrats with this nice piece of work!

Reviewer #3:

Remarks to the Author:

Summary

This manuscript describes an analysis of the effect of jet stream position on European beech growth and productivity. The title of the study is broader than the content of the study. The study results are interesting and may help to decipher the growth and productivity of European beech forests. However, I consider it a significant shortcoming of the study that the authors only present and discuss the proportions of explained variance. A discussion of the proportion of unexplained variance, which dominates the analysis by far, does not take place, which significantly reduces the significance of the results. The authors also do not use technical terms consistently throughout, which considerably lowers the manuscript's readability. The following is a list of comments I have on the manuscript.

General comments

- Please change the title of the manuscript. It suggests that an impact on forest growth has been studied in Europe. Based on the manuscript, it is not clear what forest type was investigated. The only clear thing is that European beech's radial growth and productivity were studied and modeled across sites in Europe. At least, I assume that based on the statement "To quantify carbon uptake by the forest ecosystems at our study sites".
- Please omit all occurrences of ", respectively" where the relationship between numerical values and context is clear. It is redundant, then.

- Omit all repeated definitions of variables, acronyms, ...
- Please use the same technical terms for the same quantities.
- Please check the entire text for missing blank characters in equations, between numbers and units, ...

Specific comments

L97: The abbreviation "JSL" is defined by "mid-latitude jet stream". This contradicts its definition in L75 as "jet stream latitude".

L131: Please provide the Latin species name for "European beech".

L133-L134: What does "at the same sites" mean? What is the spatial representativeness of tree ring chronologies vs the spatial representativeness of DGVM? Please clarify. In L614, you state that the resolution of the TRENDY model ensemble varies from 0.5° to 2°. What is the meaning of "site" in the context of this resolution of about 50 to 200 km? Please provide more information on the sampling sites of the tree rings.

L145-L152: Please specify in which years the jsIPC1 and jsIPC2 extremes occurred in the text.

L159: What is the meaning of "wintertime 25"? Is this a reference to the definition of "wintertime"? Please clarify.

L172-L184:

- Please provide long names for the abbreviations and acronyms used in the figure caption.
- I suggest deleting supplementary information in parentheses (except symbols, abbreviations, acronyms). Either information is essential and should be written without parentheses, or it is unimportant and can be omitted.
- Please replace "surface temperature" with "surface air temperature". Surface temperature is the temperature of the earth surface. I assume that you mean the temperature of the near-surface air layer.
- Please replace "500-mbar" with "500 mbar".
- Please include all missing commas, i.e., a, d, g, j ...

L191-L194: I suggest mentioning "Scandinavian blocking weather" first, instead of mentioning Scandinavia in parentheses as supplementary information.

L205: Please replace "beech" with "European beech".

L208-L215:

- What do you mean by "tree growth" and "growth"? Please be more specific. Do you mean radial growth, height growth, ...? What is the difference to "forest growth" mentioned in L203?
- The heading of this paragraph includes "forest growth and productivity". Is "forest growth" and "tree growth" that same analyzed quantity?

L223: Please uppercase "central" and "northern".

L246:

- Figure 2 is small. I suggest deleting the redundant axis labels to enlarge the drawing areas of the subplots.
- Results from TRW and DGVM are both given as dots. How do I have to interpret these dots for DVGM? I assume that the TRENDY model results are available as gridded data. How did you produce this "site-specific" information from the gridded data? Please provide more details.

L249: Delete all uninformative text parts in parentheses or delete the parentheses. Where do I find "STable1"?

L249, Table S1: How are JSL, heatwaves and drought events connected (see L117-L119)? In Table S1, I neither find the years 1976 and 2003 in which extreme meteorological conditions dominated

the European weather nor do I find a discussion on the lack of these years. How many heatwaves and drought events are not detected by the proposed approach? If I look into <https://doi.org/10.1016/j.ejrh.2015.01.001>, I find many years that are not indicated as "common extremes to JSL and TRW", although large parts of Europe were affected. Please discuss this issue.

L252: What do you mean by "tight"? Please be specific. By looking at the legend values of Fig. 3, "tight" is relative. The correlation coefficient values vary from -0.3 to 0.3, which is low. Are the correlation coefficient values significant? Please provide information on the significance of the correlation coefficient values.

L257-L258: Is "tree growth" equivalent to "forest growth"? Please replace "tree-growth" with "tree growth".

L257-L266: Given the explained variance of 29%, how many significant tree growth modes did you find in total? This information would be interesting to the readership for judging the importance of the remaining unexplained 71% of variance and noise in the site-specific tree growth data.

L277: What is the meaning of "common period"? Please clarify.

L281-L286: Please be more specific: What was simulated by LMM? "Forest growth", "beech growth", or both? This is not clear from these lines.

L283-L290: What about the significant shares of unexplained variance? I miss a discussion on the unexplained variance.

L294-L295: What is "beech growth"? Do you mean "radial growth of European beech"? Please clarify and change accordingly. Here, and everywhere else in the text where necessary.

L333-L337: Please be more specific concerning the target variables presented in this figure. Is there a difference between "observed beech radial growth" and "tree growth"? Interpreting these technical terms, there must be a difference. What did you simulate using LMM? What are "extremes in tree growth"? Do they relate to "observed beech radial growth" or "LMM-simulated tree growth"? This needs clarification. Here, and everywhere else in the text where necessary.

L336: Why is there a reference in the figure caption to another figure?

L337: Is there a difference of "mean (1950-2005)" (L250) to "common period (1950-2005)" (L277) and "long-term mean (1950-2005)"?

L337: What is the meaning of "expressed as percentages from the mean (1950-2005)"? Should this be the same as "percent deviations from the long-term mean (1950-2005)" (L250)?

L344-L347: This statement is not understandable for me. Please provide more information on "dominant atmospheric circulation patterns" that are not affected by the jet stream variability. The jet stream significantly determines dominant atmospheric circulation patterns in the North Atlantic-European region. It is a dominant atmospheric circulation pattern. I disagree with this statement as you have not verified and presented the spatiotemporal variability of other atmospheric circulation patterns relevant to the research question.

L345: Please delete the comma.

L349-L352: Why do you think that JSL plays a crucial role in the European dipole of carbon uptake and storage? You only provided results on the explained variance, which is relatively low. I did not find any information on the remaining unexplained variance that dominates the total explanation of variance.

L356: See comment on L249, Table S1: How are JSL, heatwaves and drought events connected (see L117-L119)? In Table S1, I neither find the years 1976 and 2003 in which extreme meteorological conditions dominated the European weather nor do I find a discussion on the lack

of these years. How many heatwaves and drought events are not detected by the proposed approach? If I look into <https://doi.org/10.1016/j.ejrh.2015.01.001>, I find many years that are not indicated as "common extremes to JSL and TRW", although large parts of Europe were affected. Please discuss this issue.

L576-L578: Why did you choose reanalysis with a resolution of 2.5 deg (approx. 250 km)? This resolution is very coarse when it is used for comparison with "tree site data". How did you create site-specific information? Please clarify and provide more details.

L582: Please be more precise and replace "temperature" with "air temperature".

L611-L625: Please provide substantially more information on the following:

- For which "forests" was carbon uptake quantified? For European beech forests?
- What does "forests" mean in the context of this study. Please provide more precise information on the forest ecosystems that were included in the study.
- Where are the forests located?
- At what elevation are the forests located? Are there sufficient beech chronologies for the highest elevations in the low mountain ranges and river valleys?
- How were coniferous forests, mixed forests, and other deciduous forests treated?
- For what area are the European beech chronologies representative? For the entire European forest area?
- Why were only beech chronologies used for this study?

L641-L643: Did you use the Pearson correlation coefficient between 2.5 deg data and "tree site data"? Does this make sense?

L644-L645: What does it mean "we composited the set ..."? What kind of composition did you apply? Please elaborate.

L655-L656: What does it mean "we composited GPP ..."? What kind of composition did you apply? Please elaborate.

Supplementary materials

Caption Fig. S1:

- Please provide the long name for "PCA" in the figure caption. All figure captions should be understandable and unambiguous on their own.
- Please change "period common period" to "common period".
- Please homogenize the definition of "JSL". Here, it is "jet stream latitudinal position".

Caption Fig. S2:

- Please provide the long name for "TRW" in the figure caption.
- Please replace "temperature" with "air temperature".
- I cannot find the values 0.28 and 0.36 in the legend. Please provide more precise labelling of the color bar.
- Please delete "respectively". It is redundant.

Caption Fig. S3:

- Please provide the long name for "LMM" and "JSL".

REVIEWER COMMENTS

Reviewer #1 (Remarks to the Author):

In their study, Dorado-Liñán et al. analyse tree ring width (TRW) chronologies to understand how weather variability associated with the Jet Stream location (JSL) affect vegetation growth. They then combine the analysis with vegetation models to derive impacts on primary productivity due to JSL variability. The manuscript with well written and well referenced, and the topic is quite relevant, given the uncertainties associated with summer extreme events and their impacts on vegetation [1]. However, I find that the dominance and causal links of JSL variability in controlling TRW and GPP are over-emphasised and not fully supported by the results in the study as is, especially since uncertainties are treated rather superficially. I explain my concerns and make specific suggestions below.

We very much appreciate the comments of reviewer#1, whose assessment has helped us revise our presentation of the relationship between jet stream variability and radial tree growth to make this more convincing. The linear mixed effect model part of the manuscript definitely benefitted from the added details. Specifically, we have been more specific in the main text (e.g., Line 316-317; L653-654; L659-660) and completed the information in the supplementary material to solidify our analyses and justify our choices for the fixed and random effects that we tested. In the revised manuscript, we explicitly tackle the uncertainties associated with the summer JSL-European beech radial growth relationship, providing more information on the portion of variability not explained by the summer JSL and discussing its potential other drivers (e.g. L413-425, L456-489). Further details on the changes made are provided below in our answers to specific comments.

The modelling approach needs to be clarified:

The LMM model explains 46% and 33% of variability during extremes of the primary (trwPC1) and secondary (trwPC2); however, these two PCs explain, respectively 19% and 10% of TRW variance. This means that the LMM model would explain, at best, less than 10% of the variance in TWR.

We thank the reviewer for this useful comment and acknowledge that the predictands of the model were not clearly enough stated in the M&M section. The predictands of the LMM were not the primary and secondary PCs modes of tree growth trwPC1 and trwPC2, but rather the 344 individual tree-ring chronologies covering the period 1950-2005. The primary and secondary PC modes of tree growth (trwPC1, trwPC2) were used in preliminary analyses for exploring the common climatic drivers across the tree-ring network (i.e., climate drivers of the common radial tree growth variability at the continental scale, new **Figures S10** now included in SI) and to detect the years showing the largest anomalies in European beech radial growth. (i.e., extremes listed in **Table S1**) to be compared with the JSL extremes.

Thus, the selected LMM explains 32% of the interannual growth variability of the entire tree-ring width network for the period spanning 1950-2005 and up to 46% of variability during extreme years. Given the large spatial extent of our datasets, we consider these numbers to be surprisingly strong, as the two main modes of summer JSL variability alone explain almost half of the continental-scale change in European beech radial growth during years with extreme summers. The determining effect of local summer climate on forest ecosystem productivity and growth is well-known. In this paper, we quantify and extend that link using upper tropospheric zonal wind, the dynamical driver of European summer weather patterns, and thus to a spatial domain that vastly exceeds that of local surface meteorological observations.

In the revised manuscript, we now explicitly mention the predictands in the M&M section to clearly state that the model is built using the individual tree-ring chronologies of the entire network and not only the primary modes of growth variability (L653-654). We have also rephrased the corresponding part in the results section (L316-317)

Qualitatively the model seems to be able to reproduce the spatial variability of the different JSL modes, which can be because of the use of geographical fixed effects (lat, lon, elev). In fact, if I read Table S4 well, the LMM with latitude/longitude and elevation as fixed effects is the one with lowest AICc, as highlighted in bold by the authors. Was this the model used to derive Figure 4? This is not clear at all to the reader. Nevertheless, the ANOVA analysis in Table S4 seems to invalidate the usefulness of JSL in explaining variations in TRW.

Table S4 refers exclusively to the components of the random effects, i.e., slope and intercept or grouping factor. The geographical parameters are included exclusively as random slopes (they were non-significant as fixed effects, see **TableS2** where fixed effects are listed). As shown in **TableS3**, the fixed factor included in the final model (the one used to derive maps in **Figure 4**) only included three climate predictors.

We have rephrased the captions of **Table S3** and **Table S4** to state more clearly what is shown in each table: fixed and random effects. In addition, we have included the equation of the selected mixed effect model in the SI (**Equation S1**).

Finally, what were the “random effects” considered? Did the authors consider “site” as a random effect? This would be reasonable as there may be a high fraction of variance explained by site-specific factors that cannot be accounted in the model.

As shown in **TableS4**, site was considered as a grouping factor (random intercept), but considering “year” as a grouping factor led to a more parsimonious model (lower AICc), which was our main criterion for model selection.

The model using “Site” as a random intercept was also feasible, but the AICc was higher, and the explained variance lower compared to the model with “year” as random intercept.

In addition to the modification of the **Table S4** caption, we now explicitly mention in the M&M section that both site and year have been tested as grouping factors in the LMM (Line 660)

The authors focus on summer jet stream variability and concurrent impacts on TRW. The also include summer JSL from the previous year in their modelling to account for legacy

effects. It is unclear why the authors did not include winter or spring JSL variability, since most summer droughts have been shown to have an important spring rainfall deficit component [2].

We agree with the reviewer that both European beech tree growth and summer heatwaves and/or droughts might also be influenced by JSL variability in prior seasons. However, those influences are typically more local or regional, as opposed to the distribution range-wide patterns assessed herein (Muffler *et al.*, 2020). Moreover, the atmospheric circulation patterns, and thus geographical precipitation patterns related to the main mode of winter and spring JSL variability, differ quite strongly from those in summer (Belmecheri *et al.*, 2017). To study geographical patterns in tree growth variability across our European beech TRW network, we preferred not to mix the different surface climate gradients resulting from different patterns in the upper atmosphere in different seasons. We therefore focus our analysis on a single and strong common driver of European beech growth variability across its distribution range, which is summer climate (Hackett-Pain *et al.*, 2016; Muffler *et al.*, 2020). Variability of summer temperature and precipitation are known to be a major component of tree growth variability because this is the season when climate may push tree species over their physiological thresholds of tree death (Walther *et al.*, 2021). We chose the summer season because 1) European beech is known to be sensitive to summer climate across its distribution range, 2) summer climate is likely the only common driver of tree growth across the 344 forest sites in the network and 3) summer climate has the potential to trigger widespread forest growth reduction and die-off. Such an approach focusing on the summer season is widely used, particularly to study the impact of extreme climate events (i.e., Zang *et al.*, 2018; Dorado-Liñán *et al.*, 2019; Buras, Rammig and Zang, 2020). Other studies have considered an extended growing season, including spring (Senf *et al.*, 2020) or autumn (June to October, (DeSoto *et al.*, 2020)), but all of them have in common the inclusion of summer because that is indeed the pivotal component.

This being said, we agree with the referee that other seasons may also play a role. During the data exploration analyses, we conducted a correlation test that now has been included in the SI (see new **Figure S10**). In this analysis, we correlated the two main modes of network-wide TRW variability (trwPC1 and trwPC2) to monthly and seasonal JSL variability per longitudinal cell over the North Atlantic/European domain. **Fig. S10** shows that significant correlations with JSL variability in seasons other than summer was limited to only a few longitudinal cells, and that summer JSL was the only variable that correlated significantly with trwPC1 and trwPC2. In fact, the trwPC1 was largely unaffected by the JSL variability in spring or winter. Accordingly, we have included a more specific mention of the relationship between tree growth and JSL in other climatic seasons in the Discussion section (L409-425).

Regarding the inclusion of previous year's summer season, legacy effects on tree growth are very well documented in literature (see the corresponding discussion paragraph in the paper L477-489). In the case of European beech, the severity of previous summer does not only imply a potential reduction of the next year's tree growth due to reduced carbon reserves, but also to the allocation to reproduction (Hackett-Pain *et al.*, 2015, 2018; Ascoli *et al.*, 2017), which may reduce growth even more in the next year.

The comparison with GPP from DGVMs is also problematic, as GPP corresponds to assimilation, but does not necessarily translate to growth. A better metric for comparison with TRW would be NPP or changes in aboveground C veg, which should be available from model outputs. Additionally, the authors chose the overall GPP simulated by models, while they could have selected the PFT-specific output, which would be more sensible to compare with TRW from a single species. The authors explain that the choice was motivated by the fact that not all models include a PFT that can be related to European beech, but most models will include a temperate deciduous PFT. Refining the analysis by using PFT-specific GPP (or rather NPP as suggested above) might improve the comparison with results for TRW in Figure 4.

We appreciate the reviewer's suggestions for a more detailed consideration of the DGVM output. We agree that PFT-specific fluxes are preferable over generalized fluxes, despite the lower number of models that provide them. Accordingly, we have sub-selected the eight DGVM models that have a PFT that corresponds to European beech. Reassuringly, the PFT-specific GPP results differ only minimally from the previously reported overall fluxes (*Figure A*).

Figure A. Anomalies in observed GPP during summer JSL extremes i.e., D90 and D10 of jslPC1 and jslPC2 scores (see Table S1) considering all plant functional types (PFT, upper panel) and only the models containing a specific European beech PFT (lower panel). Anomalies are expressed as percentages from the mean for the period 1950-2005.

Yet, using a temperate deciduous PFT increases the accuracy of the comparison between modelled GPP and radial tree growth. *Figure 2* and *4* have been modified and now display GPP information related to temperate deciduous PFT. Regarding the use of GPP vs. NPP, we see the referee's point that NPP is conceptually closer to radial tree growth than GPP. However, the way NPP is implemented in the TRENDY DGVMs (as a fixed or dynamic fraction of GPP), we doubt that using NPP rather than GPP itself would provide an advantage. On the contrary, it may even introduce more uncertainties into our comparisons, because it is an imperfect derivative of GPP: NPP is determined directly from the difference between photosynthesis (GPP) and respiratory losses, with no explicit representation of growth processes. Tree growth is highly sensitive to different

environmental drivers than photosynthesis (particularly soil water) and may even exert some control over photosynthesis through internal feedbacks on the leaf area that can be hydraulically sustained. Importantly, this means that anomalies in simulated NPP may not directly translate into growth anomalies. This is particularly relevant given the lack of representation of sink processes in the models (e.g., Zuidema, Poulter and Frank, 2018; Fatichi *et al.*, 2019; Friend *et al.*, 2019).

Despite these caveats, we have downloaded and processed the NPP output from the TRENDY models. As expected, the results for NPP (**Figure B**) show similar spatial patterns of positive-negative anomalies across the network as those obtained from GPP. However, a small number of sites display unrealistic anomalies of +400% of NPP change (see middle panel). These sites are primarily located at high elevation in Italy and may miss accurate environmental constraints on carbon allocation to NPP. Based on these considerations, we remain convinced that GPP is the most reliable DGVM output and that its comparison with tree growth allows for an important interpretation towards biomass production efficiency (i.e., the fraction of GPP that is allocated to structural biomass in a given year). We thus prefer to keep the GPP results in the main manuscript, but would add those for NPP to the supplementary material at the editor’s discretion.

Figure B. NPP and GPP during summer JSL extremes. Anomalies in observed European beech NPP (upper and middle panels) and GPP (bottom panel) during summer JSL extremes i.e., D90 and D10 of jsIPC1 and jsIPC2 scores (see Table S1). Anomalies are expressed as percentages from the mean for the period 1950-2005. Upper panel shows NPP anomalies for a common scale of change (-100% to +100%). Middle panel shows the full range of anomalies of the network.

Finally, in the abstract, the authors state that JSL “provides a synthetic and robust physical framework that integrates climate variability not accounted by atmospheric circulation patterns”. After this, one would expect a comparison with traditional atmospheric circulation indices, which have also shown strong relationships with weather variability and vegetation productivity in Europe, such as the NAO, East-Atlantic or Scandinavian Patterns [3], [4], including their winter phases.

We fully understand the reflection of the reviewer on the exploration of the main atmospheric circulation patterns. As indicated by the reviewer, it is already well-known that the main atmospheric circulation patterns are linked to the North Atlantic jet stream (e.g. Woollings, 2010; Madonna *et al.*, 2017). In particular, changes in the jet stream latitude over the North Atlantic/European realm are usually connected to specific phases of the NAO and EA (Woollings *et al.* 2010, Woollings and Blackburn, 2012). Following the reviewer's suggestion, we include a new figure showing the relationship between the modes of variability of European beech tree growth and the variability of the summer North Atlantic Oscillation and the East Atlantic pattern (**Fig. S9**). As expected, the results show that the first and second modes of tree growth variability are linked to the dominant teleconnection patterns over Europe, which are also related to the modes of JSL variability.

However, some portion of JSL variability is not captured by specific phases of the prominent atmospheric circulation patterns and specifically their indices (Fyfe and Lorenz, 2005; Monahan and Fyfe, 2006; Madonna *et al.*, 2017). It is that portion which motivated us to conduct this study on JSL impacts on terrestrial ecosystems. Furthermore, the jet stream is a real and measurable physical component of the climate system, whereas NAO and EA are statistical representations of the atmospheric state. Finally, whereas the behaviour of the jet stream under future anthropogenic climate change is a topic of much debate and ongoing research among climatologists (e.g., Screen and Simmonds, 2014; Shepherd, 2014; Francis and Vavrus, 2015; Peings *et al.*, 2018), this is not necessarily the case for traditional atmospheric circulation indices. We therefore regard the JSL as a more timely, comprehensive, and tangible framework to study climate-biosphere interactions.

We did not include the dominant atmospheric circulation patterns (NAO or EA index) in our model of network-wide radial tree growth, because they are related to our JSL indices (**Fig. S9**), will provide redundant information, and likely lead to multicollinearity and overfitting in the model. However, we agree with reviewer#1 that it is relevant to clearly state the relation between tree growth, JSL, and the summer NAO and EA. We have therefore included a time series comparison between the main modes of JSL variability and tree-growth variability with July-August NAO and EA, displaying the significant correlation among the different time series. This is now included in the SI (new **Figure S9**) and in a brief discussion in the main manuscript (L393-403)

Figure 2 and corresponding discussion: I wonder how significant are differences between composites in TRW and GPP both per site (analogue to the stippling in Figure 1) and in space. specifically, jsIPC1_NW and jsIPC2_SE look spatially very similar, which might be explained by the fact that Temp and Precip anomalies for these two composites are roughly similar.

We have now included the significance in **Figure 2**, **Figure 4** and **Figure S6**, using the same methodology that we used in **Figure 1**: we computed significance of site anomalies by Monte-Carlo tests with 1000 permutations (Methods: L639-640, L650-651)

In addition to this, the variability in the strength (significance) of the influence of summer JSL on radial tree growth and GPP across the distribution range has been explicitly mentioned throughout the description of the results (L218-252).

The correlation maps between TRW and Temp/Precip in Figure S2 are much more heterogeneous than the patterns in Figure 2. Is it possible that other variables, for example radiation and cloudiness [5] could be a predominant factor in higher latitude / energy limited regions? This could also explain with correlations in most sites are rather low (a correlation of 0.28, even if significant, is quite weak, it corresponds to less than 10% of variance explained!).

We thank the reviewer for this useful comment. With **Figure S2**, we aimed to show a general picture of the summer climate constraint on European beech radial growth across the 344 sites: a generalized negative effect of temperature and a variable dependency on precipitation.

Fig. S2 is therefore not intended to show the main climate driver at every site but a general overview of the season under study. Of course, other variables such as the surface solar radiation/percentage of cloud cover may also play a role in driving local population-level radial tree growth.

As suggested by the reviewer, we have tested potential correlations with summer surface solar radiation and percentage of cloud cover in the initial data exploration, because of the existing sensitivity of European beech to surface solar radiation (Dorado-Liñán *et al.*, 2017). We have redone **Fig. S2** with EOBS data (0.25° resolution), including an additional panel showing the correlation maps of TRW and surface solar radiation. Accordingly, **Fig. S2** now shows the correlation maps of TRW with temperature, precipitation and surface solar radiation, and highlights the significant correlations to give a more complete picture of the summer climatic constraints on beech growth.

Correlation maps with cloud cover percentage (using CRU data, see **Figure C** below) rather than surface solar radiation show very similar patterns. We have not included these maps using CRU data in **Fig. S2**, since they provided redundant information.

Figure C. Pearson's correlation coefficients between tree-ring width (TRW) chronologies and July-August air temperature (a); July-August precipitation (b) and percentage of cloud cover (c) for the period of analyses 1950-2005. Climate data are derived from CRU TS4.04 gridded database 0.5° resolution (Harris *et al.*, 2020). Solid circles indicate significant correlations between the tree ring record and the corresponding meteorological record ($p < 0.05$).

Figure 3: it is not clear how one can compare panels (a-b) with (c-d) without knowing how the trwPCs relate to the jslPCs.

We have added additional text explaining this Figure (L282-285 and L296-299) and included an additional figure in SI (*Fig. S4*), showing the agreement between PC scores time series following the reviewer's suggestion.

Small comment:

Figure S1 – there is no green, rather cyan.

Changed according to reviewer's suggestion.

[1] M. Reichstein et al., "Climate extremes and the carbon cycle," *Nature*, vol. 500, no. 7462, pp. 287–295, Aug. 2013, [Online]. Available: <http://dx.doi.org/10.1038/nature12350>

[2] E. M. Fischer, S. I. Seneviratne, P. L. Vidale, D. Lüthi, and C. Schär, "Soil moisture–atmosphere interactions during the 2003 European summer heat wave," *J. Clim.*, vol. 20, no. 20, pp. 5081–5099, 2007.

[3] L. Comas-Bru and F. McDermott, "Impacts of the EA and SCA patterns on the European twentieth century NAO-winter climate relationship," *Q. J. R. Meteorol. Soc.*, 2013, doi: 10.1002/qj.2158.

[4] A. Bastos et al., "European land CO₂ sink influenced by NAO and East-Atlantic Pattern coupling," *Nat. Commun.*, vol. 7, 2016.

[5] X.-P. Song et al., "Global land change from 1982 to 2016," *Nature*, vol. 560, pp. 639–643, 2018, doi: 10.1038/s41586-018-0411-9.

Reviewer #2 (Remarks to the Author):

The paper submitted by Dorado-Liñan et al. assesses a relationship between jet stream position variability and beech growth using dendrochronology and forest productivity using a vegetation modelling approach. This is a novel and interesting test, with potentially widespread consequences for beech survival and beech forest functioning. I very much liked reading this innovative paper and think it is suitable for the readership of *Nature Communications*.

We thank reviewer #2 for the positive assessment of our manuscript. We have taken into consideration all of their helpful comments and suggestions. We believe that the revisions have improved the quality and clarity of our study and that now there is a clear story that links the different aspects of the study. Major revised parts in the manuscript are highlighted in red.

These are my comments (major and minor comments are mixed together, sorry for that):

*In the abstract: perhaps you can add a final concluding sentence focused on the consequences of your findings and/or implications for forest functioning?

We agree with the reviewer that the abstract would benefit from an additional sentence on forthcoming consequences of changes in JSL for forest functioning. Unfortunately, we do not have room for it: the abstract is already 150 words (the Nature Communications limit) and there is no suitable other sentence to be deleted instead. We do address this broader relevance of our study in the Discussion.

*L128: I would already here briefly describe the methods and data sources of the climatological part of your analyses (if you decide to keep it in the main text, but see further).

Done based on the reviewer's suggestion. See text inserted in L132-135

*L132, L600 and elsewhere: does the word chronologies reflect to one tree ring series of one individual beech tree? I think not, but this remains unclear as that information is also not available in the Methods section on L.601 where you mention that your database comprises 344 chronologies and 9402 individual tree-ring width series? How many trees were then included in the study in the end? Confusingly, on L650 you then use the term '344 TRW chronologies'. Is each circle in Fig. 2 and 3 one tree? Could you add maps/histograms/graphs of the sampled trees, the length of the timeseries available, and the year of sampling?

We apologize for the confusion. Every (site) chronology is the average of a variable number of tree-ring series sampled at a given site and represents the annual growth variability that is common to the (European beech) trees at that site. We appreciate the reviewer's suggestion and have included a new Figure in SI (*Figure SI*) and corresponding text in M&M (L584-591), with descriptive graphs of the TRW data used. The new *Fig. SI* contains a map showing the distribution range of European beech and the location of the 344 sites. It additionally shows the time-span covered by the individual tree-level tree-ring series (not chronologies) and the histograms of the distribution of site chronologies according to elevation, latitude, and longitude.

*In the discussion, no reflection is made on the potentially moderating effects of tree mixtures on the impacts of extreme events. Indeed, a growing body of literature shows that drought/heat impacts on (beech but also other species) forests are lower in multispecies forests that have a higher tree species richness (e.g. Sousa-Silva et al. 2018 *Global Change Biology*). Do you have information on the forest composition of the sampled beech forests across Europe and on co-occurring tree species? If that information is available, you could test the influence of that as a covariate in your models.

We agree with the reviewer that further site and forest composition information would allow for a more complete analysis of the factors influencing the impact of JSL on tree growth. Unfortunately, that information is not available for a relevant portion of the EBTRN database. The EBTRN database is a compilation of tree-ring data from European beech sites that were sampled for different purposes over several decades. The information that is available at all sites is generally limited to the tree-ring data.

We have considered collecting some site information (such as forest density, co-occurring tree species, etc) from Forest Inventories. However, such data is not open-access and not easily accessible in some countries/regions and we were not able to include this data in

our analyses. Further, forest inventory data from different organizational units (e.g. countries) is not always compatible (McRoberts *et al.*, 2009), which would introduce unwanted uncertainty in such an analysis.

Although we do not have information on forest composition in our database, we acknowledge that part of the unexplained variance could be due to forest management and composition, and we now explicitly mention this in the discussion (L458-462).

*I personally find the climatological part of the paper the least innovative and relatively descriptive. For instance, on L140 till L200 and in Fig. 1, it remains unclear to me whether the maps and descriptions represent single-year (or weekly/daily) examples of each typical position of the JSL? If that is the case, could you mention the date/year in which these situations occurred in Fig. 1? Without devaluing the scientific value of this climatological part, it very much reads like a ‘textbook’ text on jet stream variability. To me, the central and most innovative part is the relationship with the tree growth and forest productivity starting on L203. I understand you need the climatological introduction, but perhaps you can consider to move a large part, or even the majority of this section, to the Supplementary Information.

First, we would like to clarify that **Figure 1** and its description correspond to the composite analysis of the extreme events of each JSL mode (jslPC1_D10, jslPC1_D90, jslPC2_D10 and jslPC2_D90). These extreme events are chosen as representative of the JSL mode and used to characterize it. For clarity, we have now included a specific mention at the beginning of this part of results (L153-154). As the reviewer also acknowledges, this first analysis of the anomalous large-scale atmospheric conditions and the associated patterns of 2m air temperature and precipitation for each JSL mode is crucial to understand its impact on tree growth.

Secondly, we agree with the reviewer that this climatological part of our manuscript is descriptive and might not appear novel for some readers, particularly if they are familiar with this topic. However, we have to assume that many potential readers from a broad-audience journal may not be familiar with the relevant weather regimes and we believe that including such information in the main body of text will help them understand the story line of the paper. More importantly, our study is the first to show the geographical patterns of summer climate extremes (and particularly the summer climate dipole) associated with these main JSL modes and that these patterns are reflected in European beech radial growth. The climatological component of our study is thus relevant for the overall findings, and we believe that it deserves to be presented in sufficient detail in the main text.

*In many of the figures, there are a lot of abbreviations used that are not explained in the captions. To improve readability of the captions, I would at least always mention the whole term as well.

Following the reviewer’s suggestion, the full terms of the abbreviations have been included in every caption.

*Code availability (L437): is there a reason not to upload the code together with the data in an online repository?

The analyses in the manuscript were performed using existing functions from the R libraires listed in M&M. The custom code developed for the calculation of the jet stream position will be made available on Figshare along with the source data.

*The definition of summer does not include June (L580), in contrast to the meteorological definition of the northern hemispheric summer (JJA). Why did you also not consider June in your analyses? Was there a specific reason for this? In recent years, extreme droughts and heatwaves also occurred in June in Europe.

We thank the reviewer for this insightful comment. We based our definition of the July-August season for JSL on an earlier analysis of the seasonal coherence in JSL variables (Belmecheri *et al.*, 2017). Using a bottom-up approach, Belmecheri et al. showed that JSL variability was spatially coherent during the months July and August, but showed a less coherent pattern in June (and September).

That being said, the reviewer is right that June might also be a relevant month when talking about heatwaves and their impact on natural systems, especially given the significant correlations between the TRW network and June precipitation and temperature (Hackett-Pain *et al.*, 2016). To show this potential effect, we have now included the correlation coefficients between the main modes of the TRW in SI (**Figure S10**). **Fig. S10** shows that trwPC1 and trwPC2 were not significantly correlated with June JSL.

*L599: can you explain this subsetting procedure of the EBTRN network “in terms of spatial and temporal domain” a bit more? For instance, why does this subsetting procedure decrease your time window from 1468-2018 to 1950-2005? Is it because you omitted chronologies < 15 trees? Is this then < 15 trees within one site? Of what area?

This is an important point and we have included more detailed explanations in the M&M section to clarify the subsetting of the EBTRN and the characteristics of the chronologies included (L581-591)

Congrats with this nice piece of work!

Thank you!

Reviewer #3 (Remarks to the Author):

Summary

This manuscript describes an analysis of the effect of jet stream position on European beech growth and productivity. The title of the study is broader than the content of the study. The study results are interesting and may help to decipher the growth and productivity of European beech forests. However, I consider it a significant shortcoming of the study that the authors only present and discuss the proportions of explained variance. A discussion of the proportion of unexplained variance, which dominates the analysis by far, does not take place, which significantly reduces the significance of the results. The authors also do not use technical terms consistently throughout, which

considerably lowers the manuscript's readability. The following is a list of comments I have on the manuscript.

We want to thank reviewer #3 for the critical assessment of this manuscript, which helped us to improve its content. In addition, and following reviewer's suggestion, we have included additional graphs and considerably extended the discussion addressing the proportion of variance not explained by JSL and the potential factors responsible for them. Please find below more detailed answers to the specific comments and suggestions and the changes made in the manuscript.

General comments

- Please change the title of the manuscript. It suggests that an impact on forest growth has been studied in Europe. Based on the manuscript, it is not clear what forest type was investigated. The only clear thing is that European beech's radial growth and productivity were studied and modeled across sites in Europe. At least, I assume that based on the statement "To quantify carbon uptake by the forest ecosystems at our study sites".

As suggested by the reviewer, we have changed the title to better represent the most important findings of our study. The title now reads: "Jet stream position explains regional anomalies in European beech forest productivity and tree growth"

- Please omit all occurrences of " , respectively" where the relationship between numerical values and context is clear. It is redundant, then.

We have now removed this word where redundant.

- Omit all repeated definitions of variables, acronyms, ...

We have omitted all repeated definitions in the text. However, following other reviewers' suggestions, we have included some acronym definitions in the figure and table captions.

- Please use the same technical terms for the same quantities.

We have now made our use of units and technical terms consistent throughout the text.

- Please check the entire text for missing blank characters in equations, between numbers and units, ...

We have now checked the text and have removed unnecessary empty spaces between numbers and units, particularly in quantities referring to GPH

Specific comments

L97: The abbreviation "JSL" is defined by "mid-latitude jet stream". This contradicts its definition in L75 as "jet stream latitude".

We have now homogenized our definition of JSL throughout the text to "jet stream latitude". The "mid-latitude" specification was primarily to distinguish our study from the subtropical jet.

L131: Please provide the Latin species name for "European beech".

The species' Latin name has been included in L131-132.

L133-L134: What does “at the same sites” mean? What is the spatial representativeness of tree ring chronologies vs the spatial representativeness of DGVM? Please clarify. In L614, you state that the resolution of the TRENDY model ensemble varies from 0.5° to 2°. What is the meaning of “site” in the context of this resolution of about 50 to 200 km? Please provide more information on the sampling sites of the tree rings.

We thank the reviewer for this insightful comment. In our manuscript, “sites” refer to the coordinates of the 344 European beech forests, from which tree-ring chronologies were derived. We have included a new set of graphs as SI (*Fig. S1*) and further explanations of the definition of site and the characteristics of the tree-ring chronologies have been included in the M&M section (L578-581; L584-591). In the case of the DGVMs, we have extracted July-August GPP from the grid cells in which the site coordinates fall. The reviewer is right that the resolution of the TRENDY DGVM output is variable. As mentioned in the M&M section, some of the individual DGVMs have a relatively coarse spatial resolution and geographically close sites may associate with the same grid cell. We have now minimized this issue by selecting the models that provide a temperate deciduous PFT (see response to reviewer #1), which has eliminated all but one model with a coarse spatial resolution. Still, the spatial mismatch between field observations and gridded data products from Earth observations or vegetation models is a source of scaling uncertainty that many environmental scientists face regularly.

L145-L152: Please specify in which years the jsIPC1 and jsIPC2 extremes occurred in the text.

We agree with the reviewer that it is a good idea to specify the years of jsIPC extremes. We now explicitly mention some of the JSL extremes that coincide with extreme years in the published literature, along with the reported area affected by the extreme (L160-162; L169-171). This gives a first insight into the climate dipole that will be described thereafter according to *Fig.1*. We have further visualized the years of jsIPC1 and jsIPC2 extremes in *Fig. S2* and listed them in *Table S1*.

L159: What is the meaning of “wintertime 25”? Is this a reference to the definition of “wintertime”? Please clarify.

We have now clarified this sentence (L172)

L172-L184:

- Please provide long names for the abbreviations and acronyms used in the figure caption.

We now provide the full names in the caption.

- I suggest deleting supplementary information in parentheses (except symbols, abbreviations, acronyms). Either information is essential and should be written without parentheses, or it is unimportant and can be omitted.

We have now modified the caption according to the reviewer’s suggestion

- Please replace “surface temperature” with “surface air temperature”. Surface temperature is the temperature of the earth surface. I assume that you mean the temperature of the near-surface air layer.

We have replaced 'surface temperature' by 'surface air temperature' throughout the text.

- Please replace "500-mbar" with "500 mbar".

Replaced

- Please include all missing commas, i.e., a, d, g, j ...

Missing commas included

L191-L194: I suggest mentioning "Scandinavian blocking weather" first, instead of mentioning Scandinavia in parentheses as supplementary information.

We thank the reviewer and now mention "Scandinavian blocking weather" first

L205: Please replace "beech" with "European beech".

We have replaced "beech" by "European beech" throughout the text following the reviewer's suggestion

L208-L215:

- What do you mean by "tree growth" and "growth"? Please be more specific. Do you mean radial growth, height growth, ...? What is the difference to "forest growth" mentioned in L203?

We refer to radial tree growth. The use of "forest growth" was to express that the tree-ring width chronologies were sampled to represent radial tree growth variability at the stand level. To avoid confusion, we now consistently refer to 'radial tree growth' or 'European beech radial growth' throughout the text.

- The heading of this paragraph includes "forest growth and productivity". Is "forest growth" and "tree growth" that same analyzed quantity?

We have now changed the heading to "JSL impacts on radial tree growth and forest productivity" for clarity.

L223: Please uppercase "central" and "northern".

Changes made according to the reviewer's suggestion:

L246:

- Figure 2 is small. I suggest deleting the redundant axis labels to enlarge the drawing areas of the subplots.

We have removed redundant axis labels from figures 2 and 4 according to the reviewer's suggestion

-Results from TRW and DGVM are both given as dots. How do I have to interpret these dots for DVGM? I assume that the TRENDY model results are available as gridded data. How did you produce this "site-specific" information from the gridded data? Please provide more details.

In the case of the DGVMs, we have extracted summer (July-August) GPP from the grid cells in which the site coordinates fall. As mentioned in the M&M section, some of the

individual DGVMs have a relatively coarse spatial resolution and geographically close sites may associate with the same grid cell. We have now minimized this issue by selecting the models that provide a temperate deciduous PFT (see response to reviewer #1), which has eliminated all but one model with a coarse spatial resolution.

More details in our response to comment on L133-L134.

L249: Delete all uninformative text parts in parentheses or delete the parentheses. Where do I find “STable1”?

We have corrected *STable 1* to *TableS1* and have reorganized all parentheses.

L249, Table S1: How are JSL, heatwaves and drought events connected (see L117-L119)? In Table S1, I neither find the years 1976 and 2003 in which extreme meteorological conditions dominated the European weather nor do I find a discussion on the lack of these years. How many heatwaves and drought events are not detected by the proposed approach? If I look into <https://doi.org/10.1016/j.ejrh.2015.01.001>, I find many years that are not indicated as “common extremes to JSL and TRW”, although large parts of Europe were affected. Please discuss this issue.

We appreciate this valuable comment. As described by Smith (2011), there are two types of extreme events: the climatic extremes and the ecological extremes and they do not always coincide. Heatwaves, on the other hand, can be caused by several mechanisms, not only JSL extremes. In this paper, we focus on dynamically driven extreme events. This sets up a clear difference with the paper of Spinoni *et al.*, (2015), which describes meteorological and hydrological droughts, defined as shortage of precipitation over 3 to 12 months, which is a longer time-window than the one we consider in this study (2 months).

The connection between JSL and extreme weather events in Europe occurs because JSL over the North Atlantic-European domain is linked to the Atlantic storm tracks and the occurrence of persistent and strong anticyclonic anomalies that disrupt the westerly air flow named “atmospheric blocking”. Such atmospheric blocking can advect warm air to certain regions, leading to summer heatwaves. They also prevent vertical air movement, and the combination of both effects can result in drought that may affect large areas. A similar brief explanation is given in the introduction section (L102-106).

A more detailed explanation about the connection between upper-tropospheric level extremes and near-surface weather extremes can be found further below, in the answer to the comment to the line L356.

L252: What do you mean by “tight”? Please be specific. By looking at the legend values of Fig. 3, “tight” is relative. The correlation coefficient values vary from -0.3 to 0.3, which is low. Are the correlation coefficient values significant? Please provide information on the significance of the correlation coefficient values.

We have now plotted the significance of the correlation coefficients in *Fig. 2*. In addition, we have included an additional figure in SI (*Fig. S4*) where the temporal agreement between the jsIPCs and trwPCs is shown. Additionally, the adjective “tight” has been removed from the sentence.

L257-L258: Is “tree growth” equivalent to “forest growth”? Please replace “tree-growth” with “tree growth”.

We now consistently refer to ‘radial tree growth’ and ‘European beech radial growth’ throughout the text.

L257-L266: Given the explained variance of 29%, how many significant tree growth modes did you find in total? This information would be interesting to the readership for judging the importance of the remaining unexplained 71% of variance and noise in the site-specific tree growth data.

Following the reviewer#3 suggestion, we have included further information on the portion of unexplained variance by the JSL. The modes of tree growth were only used for exploring the common climate drivers of the tree-ring network (i.e., climate drivers driving the common radial tree growth variability at the continental scale, new *Fig. S10* included in SI) and to detect the years showing the largest anomalies in European beech radial growth (i.e., extremes listed in *Table S1*). Only the first and the second modes were used in these exploration analyses because they explained the most common variability. The subsequent modes explain 7% (PC3) and 5% (PC4) of common variability and a declining percentage for subsequent modes (information included now in L298-299). For the purpose of exploring the common climatic drivers of the tree-ring network and to detect the years showing the largest anomalies in European beech radial growth, the first two modes explaining the largest portion of variability were sufficient, regardless of the significance of the subsequent modes. Still, we account for common variability not included in trwPC1 and trwPC2 (i.e., variability included in subsequent PC modes) in our analyses, because the predictands of the LMM model are the 344 individual tree-ring chronologies and not the two primary PC modes.

L277: What is the meaning of “common period”? Please clarify.

The common period refers to the time period covered by all-time series (including TRW chronologies and JSL). We have rephrased this sentence as follows “Anomalies are expressed as percentages from the mean for the period 1950-2005” and have also rephrased it in the captions.

L281-L286: Please be more specific: What was simulated by LMM? “Forest growth”, “beech growth”, or both? This is not clear from these lines.

Following the reviewer’s suggestion, we have homogenized our references to beech growth/forest growth throughout the text (including in these lines) and we now consistently refer to ‘European beech radial growth’ and ‘radial tree growth’.

L283-L290: What about the significant shares of unexplained variance? I miss a discussion on the unexplained variance.

We thank the reviewer for this useful comment and now include more information on the part of variability not explained by the JSL in the discussion (L409-425). We also discuss further differences in forest characteristics across the distribution range, which are part of the non-explained variance (L456-489).

L294-L295: What is “beech growth”? Do you mean “radial growth of European beech”? Please clarify and change accordingly. Here, and everywhere else in the text where necessary.

Following the reviewer’s suggestion, we have homogenized our references to beech growth/forest growth throughout the text (including in these lines) and we now consistently refer to ‘European beech radial growth’ and ‘radial tree growth’.

L333-L337: Please be more specific concerning the target variables presented in this figure. Is there a difference between “observed beech radial growth” and “tree growth”? Interpreting these technical terms, there must be a difference. What did you simulate using LMM? What are “extremes in tree growth”? Do they relate to “observed beech radial growth” or “LMM-simulated tree growth”? This needs clarification. Here, and everywhere else in the text where necessary.

We have now rewritten the caption of *Figure 4* in order to clarify the terms we use in the table. ‘Observed beech radial growth’ have been changed to ‘radial tree growth’; ‘LMM-simulated tree growth’ has been changed to ‘simulated anomalies in radial tree growth by the LMM’ and ‘extremes in tree growth’ has been changed to ‘for the years showing the largest radial tree growth anomalies (i.e., D90 and D10 of trwPC1 and trwPC2 scores, see *Table S1, Fig. 3*). Similar changes have been applied in caption of *Fig. 2*.

We have further specified in the text that LMM simulates the radial tree growth at the 344 European beech forest sites (L316-317; L653-654)

L336: Why is there a reference in the figure caption to another figure?

We refer to *Fig. 3* in the caption of this Figure, because we believe that it helps the reader to keep track of the many analyses and to find additional information if they need it to interpret the figure (e.g., the extreme years considered for the composite). We have left the reference to *Fig. 3* in the revised manuscript, but can remove it at the discretion of the editor.

L337: Is there a difference of “mean (1950-2005)” (L250) to “common period (1950-2005)” (L277) and “long-term mean (1950-2005)”?

We thank the reviewer for pointing out this discrepancy in our wording. We refer to the same common period (1950-2005) in all cases and have homogenized the term to “mean for the period 1950-2005” throughout the text.

L337: What is the meaning of “expressed as percentages from the mean (1950-2005)” (L250)? Should this be the same as “percent deviations from the long-term mean (1950-2005)” (L277)?

These two sentences indeed mean the same thing and we have changed it to “Anomalies are expressed as percentages from the mean for the period 1950-2005” in both L273-74 and L388-389.

L344-L347: This statement is not understandable for me. Please provide more information on “dominant atmospheric circulation patterns” that are not affected by the jet stream variability. The jet stream significantly determines dominant atmospheric circulation patterns in the North Atlantic-European region. It is a dominant atmospheric

circulation pattern. I disagree with this statement as you have not verified and presented the spatiotemporal variability of other atmospheric circulation patterns relevant to the research question.

We rephrase this sentence as follows to be more specific in this matter: “Previous studies showed that jet stream movements, however, integrate spatiotemporal variability not accounted for by the main variability modes of the atmospheric circulation alone”, indicating that atmospheric circulation patterns are indeed affected by JSL variability, but cannot explain all of its variability. That statement is supported by the findings reported in the two papers cited at the end of the sentence (Woollings, Hannachi and Hoskins, 2010; Madonna *et al.*, 2017).

To further illustrate the relation between jet stream variability and the main variability modes of the summer atmospheric circulation over the North Atlantic/Europe, such as the summer NAO and the summer EA, we have added a figure in the supplementary material (**Fig. S9**) in which we show the relationship among JSL, teleconnection patterns and tree growth as well as the corresponding text (L396-399)

L345: Please delete the comma.

Deleted

L349-L352: Why do you think that JSL plays a crucial role in the European dipole of carbon uptake and storage? You only provided results on the explained variance, which is relatively low. I did not find any information on the remaining unexplained variance that dominates the total explanation of variance.

Large-scale radial tree growth/GPP patterns such as the continental-scale dipole described in our paper, can only be triggered by large-scale drivers, such as atmospheric circulation. We find that summer JSL is responsible for roughly 30% of the common variability of 344 European beech tree ring chronologies during the period 1950-2005 and is thus clearly a relevant common driver. This relevance is not only related to the percentage of explained variance - local factors may explain a larger portion of variance at most of the sites due to the contrasting abiotic and biotic conditions across the entire distribution range of the species - but to the large-scale impact that changes in that dipole pattern may have on the carbon budget. An anomaly in the summer JSL influences the radial tree growth/GPP dipole across Europe by driving large-scale climate anomalies. These climate anomalies outweigh the influence of local factors and thus leave an imprint in tree growth and ecosystem carbon uptake at the continental scale. An anomaly in local factors (i.e., those responsible for the percentage of unexplained variance, such as microclimatic conditions, forest compositions and density, disturbance dynamics etc.), on the other hand, will only have a local impact and are not visible on a continental or hemispherical scale.

We would further like to point out that the part of common variability in European beech radial growth that is explained by JSL variability may seem modest (32%) but indeed, it is not. A super fine-tuned chronology for climate reconstruction may explain 50-60% of the variance in the main driver in the best-case scenario. And this is achieved because the site has been very carefully selected to be sensitive to that main driver. The fact that a diverse network extending over most of a continent explains +30% of variability is not

obvious. On the contrary, it's quite impressive and even more when JSL variability can explain up to 46% of the growth anomaly during years of extreme summer climate. Thus, summer JSL variability alone may have substantial implications for the European carbon budget.

Following the suggestion of reviewer#1 and #3 we have included additional information on the explained and unexplained variance by JSL. See text L409-425 and L456-489.

L356: See comment on L249, Table S1: How are JSL, heatwaves and drought events connected (see L117-L119)? In Table S1, I neither find the years 1976 and 2003 in which extreme meteorological conditions dominated the European weather nor do I find a discussion on the lack of these years. How many heatwaves and drought events are not detected by the proposed approach? If I look into <https://doi.org/10.1016/j.ejrh.2015.01.001>, I find many years that are not indicated as "common extremes to JSL and TRW", although large parts of Europe were affected. Please discuss this issue.

The reviewer raises an important point, even though our study was not focused on detecting European droughts and heatwaves. We do look at the most extreme summer JSL positions to relate JSL variability to anomalies in surface weather, such as droughts and heatwaves, and then to European beech radial growth and productivity (e.g., **Fig. 3**).

The reviewer is correct that not all European summer weather extremes can be explained by JSL anomalies alone. We emphasize in our manuscript the dipole conditions that are created by extreme JSL positions (e.g., **Fig. 1**). Because of their dipole character, extreme JSL positions do not necessarily give rise to the largest or most severe summer weather anomalies. Indeed, several studies such as Zscheischler *et al.* (2018) and Schumacher *et al.* (2019) have shown that most high-impact extreme weather events occur due to a combination of various interacting physical processes. They are therefore rarely the result of the influence of a single driver such as the jet stream. For instance, the mega-heatwave of 2003 could not have happened without important anomalous conditions during the preceding seasons, such as anomalously high sea surface temperatures since May (Black *et al.*, 2004; Della-Marta *et al.*, 2007) and anomalously strong negative precipitation and soil moisture anomalies over central Europe since February 2003 (Della-Marta *et al.*, 2007; García-Herrera *et al.*, 2010). Furthermore, some of those drivers that contribute to the occurrence of summer weather extremes are related to smaller-scale spatial processes or specific to lower atmospheric levels.

The influence of lower tropospheric conditions, in addition to differing time windows (3 months vs 2 months) can therefore explain the lack of coincidence between the extremes in our study and those reported in Spinoni *et al.* (2015). The characterization of extreme events in Spinoni *et al.* (2015) is based on surface weather, more specifically on the drought indices SPI, SPEI, and RDI. The magnitude of the extremes in surface weather is not always directly proportional to the degree of anomaly of the upper-tropospheric circulation. This is the same reason, why not all the extremes in European beech radial growth coincide with extremes in the JSL: climatological thresholds do not always coincide with ecological thresholds (Smith, 2011). Indeed, 2003 is not a main European beech radial growth anomaly in our network.

Despite all of this, we can expect some agreement among heatwaves and droughts, JSL anomalies, and tree growth anomalies. Indeed, we find that most JSL extremes anomalies coincide with extreme anomalies in European beech radial growth (e.g., *Table S1, Figs. 3,4*).

We thank the reviewer for pointing out the relevance of this discussion and have now added a new paragraph that discusses the limitations of our approach to detect some summer climate extremes described in other publications, such as the heatwave of 2003. We further discuss those limitations in the discussion section (L444-455)

L576-L578: Why did you choose reanalysis with a resolution of 2.5 deg (approx. 250 km)? This resolution is very coarse when it is used for comparison with “tree site data”. How did you create site-specific information? Please clarify and provide more details.

For calibration and comparison purposes, we need long (50+year) time series of u-wind speed at 300 mbar. Such data are only available in reanalysis products. We chose the NCEP/NCAR reanalysis product based on times-series length, on reliability of the product, and on consistency with previous studies. This global reanalysis has been extensively used in studies dealing with European atmospheric blocking and Atlantic jet stream variability (e.g., Davini and D’Andrea, 2016), which enables comparisons across studies. At the time of our analyses, other reanalysis products with a finer spatial resolution were only available for shorter time periods, from 1958 (JRA-55) or 1979 onwards (i.e., ERA-Interim). They therefore did not cover the common period of our tree-ring record. Very recently, ERA5 data (previous ERA-Interim) reaching back to 1950 have been made available and we expect that a critical body of literature will make use of this data set soon.

In this study we are not dealing with site-specific information, but with large atmospheric data to describe large-scale and upper-tropospheric patterns such as the location of the jet or persistent anticyclonic anomalies. For this purpose, the 2.5 degree resolution of the NCEP/NCAR data used is more than sufficient.

L582: Please be more precise and replace “temperature” with “air temperature”.

Following the reviewer’s suggestion, we have replaced “temperature” with “air temperature” throughout the manuscript and SI.

L611-L625: Please provide substantially more information on the following:

- For which “forests” was carbon uptake quantified? For European beech forests?

We quantified both radial tree growth for European beech forests and GPP for ‘temperate broadleaf deciduous’ plant functional type which is the group in which European beech forests falls into. This has been further clarified in M&M (L600-603).

- What does "forests" mean in the context of this study. Please provide more precise information on the forest ecosystems that were included in the study.

We defined ‘forest’ broadly in this paper and include all forested areas dominated by European beech trees. We now include this information explicitly in L578-581. We further describe in the M&M that the network includes forests with different management

histories (see L588-591). Additionally, the *Figure S1* now provides more information on the geographical gradient cover by the network.

- Where are the forests located?

This is an important point. We now include clear information in *Fig. S1* on the location and the geographical gradient covered by the European beech tree-ring network. The metadata will be uploaded at a public repository, so further details will be accessible to the readers upon publication.

- At what elevation are the forests located? Are there sufficient beech chronologies for the highest elevations in the low mountain ranges and river valleys?

We included an additional figure in the SI (*Fig. S1*) that shows the distribution and characteristics of the chronologies included and the sites where they were sampled. As can be seen in the histograms, the network covers the entire geographical distribution of European beech in Europe.

- How were coniferous forests, mixed forests, and other deciduous forests treated?

The GPP output of the DGVMs that we used only considers the temperate deciduous PFT, so coniferous forest and other deciduous forest (e.g. boreal larch species) are not considered in our analyses. We have now added this info in L600-603.

- For what area are the European beech chronologies representative? For the entire European forest area?

We now show in *Fig. S1* that our network spans most of the distribution range of European beech and we provide clear information on the location and the geographical gradient covered by the network.

We also included further information in a paragraph related to the tree-ring width chronologies (L581-589).

- Why were only beech chronologies used for this study?

In this study, we only considered European Beech chronologies for three main reasons: the first one is that European beech is highly sensitive to extreme summer weather across its distribution range. This distribution-wide sensitivity to summer climate conditions was a must for our approach. The second reason is that the distribution of European beech covers most of the European continent (*Fig. S1*), has a rather continuous distribution range, and is considered an ecologically and economically important species in Europe. Thirdly, the European Beech Tree-ring Network is the only single-species network with such dense spatial and temporal coverage of chronologies that represents the full variability and distribution of the tree species.

L641-L643: Did you use the Pearson correlation coefficient between 2.5 deg data and “tree site data”? Does this make sense?

We calculated correlation coefficients between the tree-ring chronologies and the scores of the first and second mode of JSL variability (see lines L630-L632). The JSL variability modes are the ones calculated based on 2.5° data, but represent patterns that work on much larger spatial scales. The motivation behind these analyses was to approximate an

Empirical Orthogonal Function (EOF), a representation commonly used in climatology to study possible spatial modes of variability, such as the dipole we study in this paper. With this particular purpose, the analyses make sense.

L644-L645: What does it mean “we composited the set ...”? What kind of composition did you apply? Please elaborate.

L655-L656: What does it mean “we composited GPP ...”? What kind of composition did you apply? Please elaborate.

Following the suggestions by reviewers #2 and #3, we now more clearly explain the composite maps and what they represent (L637-639, L648-649).

Supplementary materials

Caption Fig. S1:

- Please provide the long name for “PCA” in the figure caption. All figure captions should be understandable and unambiguous on their own.

We now provide the long name for “PCA” in the caption of *Table S1* and *Fig. S2* (former Fig. S1)

- Please change “period common period” to “common period”.

changed

- Please homogenize the definition of “JSL”. Here, it is “jet stream latitudinal position”.

We have now homogenized our definition of JSL throughout the text to “jet stream latitude”.

Caption Fig. S2:

- Please provide the long name for “TRW” in the figure caption.

- Please replace “temperature” with “air temperature”.

- I cannot find the values 0.28 and 0.36 in the legend. Please provide more precise labelling of the color bar.

- Please delete “respectively”. It is redundant.

We have made the changes to *Fig. S2* that the reviewer suggests and now we show the significance of the correlations in *Fig. S2*.

Caption Fig. S3:

- Please provide the long name for “LMM” and “JSL”.

We have now included full names in the caption

References

Ascoli, D. *et al.* (2017) ‘Inter-annual and decadal changes in teleconnections drive continental-scale synchronization of tree reproduction’, *Nature Communications*, 8(1), pp. 1–9. doi: 10.1038/s41467-017-02348-9.

- Belmecheri, S. *et al.* (2017) 'Northern Hemisphere jet stream position indices as diagnostic tools for climate and ecosystem dynamics', *Earth Interactions*, 21(8), pp. 1–23. doi: 10.1175/EI-D-16-0023.1.
- Black, E. *et al.* (2004) 'Factors contributing to the summer 2003 European heatwave', *Weather*, 59(8), pp. 217–223. doi: <https://doi.org/10.1256/wea.74.04>.
- Buras, A., Rammig, A. and Zang, C. S. (2020) 'Quantifying impacts of the 2018 drought on European ecosystems in comparison to 2003', *Biogeosciences*, 17(6), pp. 1655–1672. doi: 10.5194/bg-17-1655-2020.
- Davini, P. and D'Andrea, F. (2016) 'Northern Hemisphere Atmospheric Blocking Representation in Global Climate Models: Twenty Years of Improvements?', *Journal of Climate*, 29(24), pp. 8823–8840. doi: 10.1175/JCLI-D-16-0242.1.
- Della-Marta, P. M. *et al.* (2007) 'Summer heat waves over western Europe 1880–2003, their relationship to large-scale forcings and predictability', *Climate Dynamics*, 29(2), pp. 251–275. doi: 10.1007/s00382-007-0233-1.
- DeSoto, L. *et al.* (2020) 'Low growth resilience to drought is related to future mortality risk in trees', *Nature Communications*, 11(1), p. 545. doi: 10.1038/s41467-020-14300-5.
- Dorado-Liñán, I. *et al.* (2017) 'Coexistence in the Mediterranean-Temperate transitional border: Multi-century dynamics of a mixed old-growth forest under global change', *Dendrochronologia*, 44, pp. 48–57. doi: 10.1016/j.dendro.2017.03.007.
- Dorado-Liñán, I. *et al.* (2019) 'Geographical adaptation prevails over species-specific determinism in trees' vulnerability to climate change at Mediterranean rear-edge forests', *Global Change Biology*, 25(4), pp. 1296–1314. doi: 10.1111/gcb.14544.
- Fatichi, S. *et al.* (2019) 'Modelling carbon sources and sinks in terrestrial vegetation', *New Phytologist*, 221(2), pp. 652–668. doi: <https://doi.org/10.1111/nph.15451>.
- Francis, J. A. and Vavrus, S. J. (2015) 'Evidence for a wavier jet stream in response to rapid Arctic warming', *Environmental Research Letters*, 10(1), p. 14005. doi: 10.1088/1748-9326/10/1/014005.
- Friend, A. D. *et al.* (2019) 'On the need to consider wood formation processes in global vegetation models and a suggested approach', *Annals of Forest Science*, 76(2), p. 49. doi: 10.1007/s13595-019-0819-x.
- Fyfe, J. C. and Lorenz, D. J. (2005) 'Characterizing Midlatitude Jet Variability: Lessons from a Simple GCM', *Journal of Climate*, 18(16), pp. 3400–3404. doi: 10.1175/JCLI3486.1.
- García-Herrera, R. *et al.* (2010) 'A Review of the European Summer Heat Wave of 2003', *Critical Reviews in Environmental Science and Technology*, 40(4), pp. 267–306. doi: 10.1080/10643380802238137.
- Hackett-Pain, A. J. *et al.* (2015) 'The influence of masting phenomenon on growth–climate relationships in trees: explaining the influence of previous summers' climate on ring width', *Tree Physiology*, 35(3), pp. 319–330. doi: 10.1093/treephys/tpv007.
- Hackett-Pain, A. J. *et al.* (2016) 'Consistent limitation of growth by high temperature and low precipitation from range core to southern edge of European beech indicates

- widespread vulnerability to changing climate’, *European Journal of Forest Research*, 135(5), pp. 897–909. doi: 10.1007/s10342-016-0982-7.
- Hackett-Pain, A. J. *et al.* (2018) ‘Climatically controlled reproduction drives interannual growth variability in a temperate tree species’, *Ecology Letters*, 21(12), pp. 1833–1844. doi: 10.1111/ele.13158.
- Harris, I. *et al.* (2020) ‘Version 4 of the CRU TS monthly high-resolution gridded multivariate climate dataset’, *Scientific Data*, 7(1), p. 109. doi: 10.1038/s41597-020-0453-3.
- Madonna, E. *et al.* (2017) ‘The link between eddy-driven jet variability and weather regimes in the North Atlantic-European sector’, *Quarterly Journal of the Royal Meteorological Society*, 143(708), pp. 2960–2972. doi: 10.1002/qj.3155.
- McRoberts, R. E. *et al.* (2009) ‘Harmonizing National Forest Inventories’, *Journal of Forestry*, 107(4), pp. 179–187. doi: 10.1093/jof/107.4.179.
- Monahan, A. H. and Fyfe, J. C. (2006) ‘On the Nature of Zonal Jet EOFs’, *Journal of Climate*, 19(24), pp. 6409–6424. doi: 10.1175/JCLI3960.1.
- Muffler, L. *et al.* (2020) ‘Lowest drought sensitivity and decreasing growth synchrony towards the dry distribution margin of European beech’, *Journal of Biogeography*, 47(9), pp. 1910–1921. doi: <https://doi.org/10.1111/jbi.13884>.
- Peings, Y. *et al.* (2018) ‘Projected squeezing of the wintertime North-Atlantic jet’, *Environmental Research Letters*, 13(7), p. 74016. doi: 10.1088/1748-9326/aacc79.
- Schumacher, D. L. *et al.* (2019) ‘Amplification of mega-heatwaves through heat torrents fuelled by upwind drought’, *Nature Geoscience*, 12(9), pp. 712–717. doi: 10.1038/s41561-019-0431-6.
- Screen, J. A. and Simmonds, I. (2014) ‘Amplified mid-latitude planetary waves favour particular regional weather extremes’, *Nature Climate Change*, 4(8), pp. 704–709. doi: 10.1038/nclimate2271.
- Senf, C. *et al.* (2020) ‘Excess forest mortality is consistently linked to drought across Europe’, *Nature Communications*, 11(1), p. 6200. doi: 10.1038/s41467-020-19924-1.
- Shepherd, T. G. (2014) ‘Atmospheric circulation as a source of uncertainty in climate change projections’, *Nature Geoscience*, 7(10), pp. 703–708. doi: 10.1038/ngeo2253.
- Smith, M. D. (2011) ‘An ecological perspective on extreme climatic events: a synthetic definition and framework to guide future research’, *Journal of Ecology*, 99(3), pp. 656–663. doi: 10.1111/j.1365-2745.2011.01798.x.
- Spinoni, J. *et al.* (2015) ‘The biggest drought events in Europe from 1950 to 2012’, *Journal of Hydrology: Regional Studies*, 3, pp. 509–524. doi: <https://doi.org/10.1016/j.ejrh.2015.01.001>.
- Trouet, V., Babst, F. and Meko, M. (2018) ‘Recent enhanced high-summer North Atlantic Jet variability emerges from three-century context’, *Nature Communications*, 9(1), pp. 1–9. doi: 10.1038/s41467-017-02699-3.
- Walthert, L. *et al.* (2021) ‘From the comfort zone to crown dieback: Sequence of physiological stress thresholds in mature European beech trees across progressive drought’, *Science of The Total Environment*, 753, p. 141792. doi:

<https://doi.org/10.1016/j.scitotenv.2020.141792>.

Woollings, T. (2010) 'Dynamical influences on European climate: An uncertain future', *Philosophical Transactions of the Royal Society A: Mathematical, Physical and Engineering Sciences*, 368(1924), pp. 3733–3756. doi: 10.1098/rsta.2010.0040.

Woollings, T., Hannachi, A. and Hoskins, B. (2010) 'Variability of the North Atlantic eddy-driven jet stream', *Quarterly Journal of the Royal Meteorological Society*, 136(649), pp. 856–868. doi: 10.1002/qj.625.

Zang, C. *et al.* (2018) 'Climate and drought responses in a continent-wide tree-ring network of European beech (*Fagus sylvatica* L.)', in *EGU General Assembly Conference Abstracts*. (EGU General Assembly Conference Abstracts), p. 13454.

Zscheischler, J. *et al.* (2018) 'Future climate risk from compound events', *Nature Climate Change*, 8(6), pp. 469–477. doi: 10.1038/s41558-018-0156-3.

Zuidema, P. A., Poulter, B. and Frank, D. C. (2018) 'A Wood Biology Agenda to Support Global Vegetation Modelling', *Trends in Plant Science*, 23(11), pp. 1006–1015. doi: <https://doi.org/10.1016/j.tplants.2018.08.003>.

Reviewers' Comments:

Reviewer #1:

Remarks to the Author:

Review of "Jet stream position explains regional anomalies in European beech forest productivity 1 and tree growth" by Dorado-Liñán et al.

I commend the authors for the thorough revision of the manuscript, which is now much improved. It is also now easier to appreciate the robustness of the results, and how this study provides a novel and innovative way to understand the impact of atmospheric dynamics on ecosystem activity. That said, I fear that some aspects still need to be clarified in order to evaluate the correctness of the statistical analysis, and – more importantly – ensure the reproducibility of the results presented here. In that regard, and for the sake of transparency, it would be advisable to publish the source code alongside the manuscript. If that is not possible, I point out below the aspects that, in my opinion, are still presented in a confusing – and unfortunately a bit sloppy sometimes – way. I also explain new concerns that emerged now that I can follow the methodology more closely.

Statistical analysis

I thank the authors for clarifying the methods, improving the tables and adding the final equation in supplement. This makes now the manuscript much easier to follow. One aspect that is still not clear to me is how the model choice is performed. My understanding from the way the methods are presented is that:

- (1) first the authors fit one model with fixed effects only, using the seven predictors listed in Table S2;
- (2) they then fit a set of models where they exclude each of the seven predictors and compare the AICc of each of these models with the first one, to estimate $\Delta AICc$;
- (3) they identify three predictors that lead to high values of $\Delta AICc$ (jsIPC1, jsIPC2, jsIPC2t) and use these as fixed effects in a new LMM and compare the AICc with a set of models including only 2 out of three predictors to confirm all three are relevant predictors;
- (4) At this stage they have a model with fixed effects only M_{FE} : $TRW \sim jsIPC1 + jsIPC2 + jsIPC2t$ (in R lme4 notation). Then they test a suite of models with jsIPC1, jsIPC2, jsIPC2t as fixed effects and additionally test for multiple random effects such as random intercepts SiteID and Year ($TRW \sim jsIPC1 + jsIPC2 + jsIPC2t + 1|SiteID$ or $1|Year$) and random slopes ($\dots + TRW|lat + TRW|lon$ etc). The different types of mixed effects tested are those described in Table 4.

I hope I understood the steps correctly. If so, I have two concerns about the methodology:

First, it is unclear to me why the authors did steps (1-3) separately to choose the fixed effects first. Rather, I would argue that a first step would be to fit models with random effects only, and test whether inclusion of fixed effects significantly improved the model fit.

Second, the authors test eight variables as random effects: jsIPC1, jsIPC2, jsIPC2t, lat, lon, ele, SiteID and Year. However, not all possible combinations of these eight variables are shown. Rather, SiteID as a random intercept is only tested for models with at least five variables as random slopes (upper row in Table S4), while more combinations are tested for random intercept Year. I have several questions/concerns in this regard:

- a) Is there a physical reason for this choice? I tried to find an explanation in the manuscript without success.
- b) Why weren't more models tested using lower degrees of freedom tested using SiteID as random intercept, as done for Year?
- c) Are the random slopes and random intercepts correlated or uncorrelated? And how is this treated in the analysis?
- d) Why isn't year considered as a potential random slope effect? In fact, this would be a key element to test, since it would allow evaluating if potential differences in TRW between years due to random variations in growth, for example related with variations in C allocation to growth vs. reproduction, could contribute to some of the variability in the data.

To improve clarity and rigor, I would recommend to rely more heavily on mathematical notation. For example, the authors can explicitly write the model equations in Tables S2-S4: $TRW = \beta_0 + \beta_1 jslPC1 + \beta_2 jslPC2 + \dots$ and so on, or at least their right-hand sides, instead of simply the names of the variables.

Thanks for explicitly showing the final model in Equation S1 now, it definitely improves clarity. But the notation is still unclear. What is γ_i and what does the i refer to? Is this TRW for each location? But then how are the random effects of elevation, latitude and longitude included in the model? The caption explains that $\mu_{i,lon,lat,ele}$ refers to random slopes varying across geographical position (latitude, longitude and elevation), but the equation specifies that m depends on i ($\mu_{i,lon,lat,ele}$). Which one is correct? Please provide full description of the different terms of the equation.

Minor comments:

$jslPC1_t$ and $jslPC2_t$ are confusing acronyms, as one could think they relate to a trend. I suggest renaming them as $jslPC1_{t-1}$ and $jslPC2_{t-1}$ to explicitly show the lag.

Line 90-91: more correctly, it is 30% of modelled gross primary productivity. Carbon uptake would usually refer to NEE. Plus, the fluxes are derived from the DGVMs only.

Line 120-122: It is still confusing that none of these two events is found in the analysis of extremes presented here. Although the authors addressed this in lines 444-455, but I do not find the explanation very convincing. In principle, the goal of using JSL is precisely to reduce the dimensionality of the drivers of TRW variability. Here is important to discuss the role of memory effects from soil drying in 2003, which has been shown to be a crucial factor to the development of the 2003 event (Fischer et al. 2007) and can likely explain why summer JSL isn't a good predictor then.

Line 661: "Starting from a saturated model, [...]" please explain what this means and clearly state what are the predictors of that model. To my understanding it is a model with all seven predictors listed in Table S2.

Table S2: my understanding is that the β , SE and p-values refer to the model with all seven predictors, while $\Delta AICc$ refers to the difference between that model and the ones excluding each predictor. This is not clear in the caption and should be stated explicitly for improved clarity and reproducibility.

Table S3: is this now showing the coefficients for a model with all three predictors, and the $\Delta AICc$ the difference between AICc between that model and the ones with 2 predictors only? Please clarify in the caption.

Table S4: VIF is not shown, why?

Fischer, E.M., Seneviratne, S.I., Vidale, P.L., Lüthi, D., Schär, C., 2007. Soil moisture-atmosphere interactions during the 2003 European summer heatwave. *J. Climate* 20, 5081-5099.

Reviewer #2:

Remarks to the Author:

I thank the authors for their thoughtful and well-elaborated responses to my earlier suggestions and remarks on the previous version of the manuscript.

I have no further comments, except for a few very small things:

*the link to the code on figshare does not work;

*I would be tempted to also include the "R code using existing functions" on the figshare repository. Also performing a linear mixed effect model requires specific choices, setting of parameters, random effects, error distribution, likelihood estimation, etc. and these are only reproducible with the full code.

I look forward to seeing this paper "in press".

Reviewer #3:

Remarks to the Author:

The authors have fully addressed my concerns. I have no further comments on the manuscript.

REVIEWER COMMENTS

Reviewer #1 (Remarks to the Author):

Review of “Jet stream position explains regional anomalies in European beech forest productivity and tree growth” by Dorado-Liñán et al.

I commend the authors for the thorough revision of the manuscript, which is now much improved. It is also now easier to appreciate the robustness of the results, and how this study provides a novel and innovative way to understand the impact of atmospheric dynamics on ecosystem activity. That said, I fear that some aspects still need to be clarified in order to evaluate the correctness of the statistical analysis, and – more importantly – ensure the reproducibility of the results presented here. In that regard, and for the sake of transparency, it would be advisable to publish the source code alongside the manuscript. If that is not possible, I point out below the aspects that, in my opinion, are still presented in a confusing – and unfortunately a bit sloppy sometimes – way. I also explain new concerns that emerged now that I can follow the methodology more closely.

We thank reviewer#1 for the positive assessment of the revised version of our manuscript. In this revision, we have further refined the explicit explanations and better justified some of the reasoning underlying our choices regarding the LMM. We are positive that the additional information provided in this new version of the manuscript will now allow for full reproducibility of the analyses. Please find our point-by-point answers below.

Statistical analysis

I thank the authors for clarifying the methods, improving the tables and adding the final equation in supplement. This makes now the manuscript much easier to follow. One aspect that is still not clear to me is how the model choice is performed. My understanding from the way the methods are presented is that:

- (1) first the authors fit one model with fixed effects only, using the seven predictors listed in Table S2;
- (2) they then fit a set of models where they exclude each of the seven predictors and compare the AICc of each of these models with the first one, to estimate ΔAICc ;
- (3) they identify three predictors that lead to high values of ΔAICc (jsIPC1, jsIPC2, jsIPC2t) and use these as fixed effects in a new LMM and compare the AICc with a set of models including only 2 out of three predictors to confirm all three are relevant predictors;
- (4) At this stage they have a model with fixed effects only MFE: $\text{TRW} \sim \text{jsIPC1} + \text{jsIPC2} + \text{jsIPC2t}$ (in R lme4 notation). Then they test a suite of models with jsIPC1, jsIPC2, jsIPC2t as fixed effects and additionally test for multiple random effects such as random intercepts SiteID and Year ($\text{TRW} \sim \text{jsIPC1} + \text{jsIPC2} + \text{jsIPC2t} + 1|\text{SiteID}$ or $1|\text{Year}$) and random slopes (... + $\text{TRW}|\text{lat}$ + $\text{TRW}|\text{lon}$ etc). The different types of mixed effects tested are those described in Table 4.

I hope I understood the steps correctly. If so, I have two concerns about the methodology:

First, it is unclear to me why the authors did steps (1-3) separately to choose the fixed effects first. Rather, I would argue that a first step would be to fit models with random effects only, and test whether inclusion of fixed effects significantly improved the model fit.

We thank the reviewer for this useful comment. We acknowledge that the steps we followed to develop the LMM needed further clarification in the M&M section. Our aim in developing the LMM was to verify the JSL-radial tree growth relationship observed in the composite maps by fitting a simple and biologically feasible model. For this purpose, the first step was to identify the main explanatory variables of tree growth, i.e., those variables that have a significant influence on radial tree growth across the distribution range (fixed effects). We selected the variables to be tested based on an exploratory analysis (see **Figures S2, S3, S4, S9 and S10**), thereby following the protocol for data exploration by Zuur et al. (2010). Once the fixed effects were selected, we added a random component to 1) control for non-independence among measurements and 2) explain part of the residual variability that, based on the dipole observed in the composite maps, we presumed was linked to the different influence of JSL according to site location and/or year. The steps we followed, defining first the explanatory variables and then modelling part of the residual variability, are a common procedure in confirmatory hypothesis testing (Barr et al., 2013). We are aware that other approaches to develop the models (including the one mentioned by the referee) might also be valid. However, we remain convinced that our approach was appropriate considering the aim of our modelling exercise, our best understanding of the ecological processes that we aim to model, and the model structure that we designed accordingly.

In the revised manuscript, we now explicitly mention the steps followed in the M&M section (L684-704).

Second, the authors test eight variables as random effects: jsIPC1, jsIPC2, jsIPC2t, lat, lon, ele, SiteID and Year. However, not all possible combinations of these eight variables are shown. Rather, SiteID as a random intercept is only tested for models with at least five variables as random slopes (upper row in Table S4), while more combinations are tested for random intercept Year. I have several questions/concerns in this regard:

a) Is there a physical reason for this choice? I tried to find an explanation in the manuscript without success.

We thank the reviewer for this useful comment. We carefully chose the random effects structure to develop an understandable and biologically meaningful model while avoiding overparameterization. In line with this goal, **Table S4** does not include all possible combinations of the random variables for two main reasons: First, our random effects' structure considered SiteID and Year only as grouping factors (random intercepts) because they represent multiple observations and thus incorporate a potential dependency among observations. Second, the other variables were considered as random slopes with either Site ID or Year as random intercepts and final model selection was based on the lowest AICc in the absence of collinearity. **Table S4** displays only a few models with Site ID as a random intercept variable in order to reduce the amount of information shown and to simplify the table (see a more detailed answer below).

Following the reviewer's suggestion, more detailed explanations on the random effects structure and combinations tested have been included in L684-704.

b) Why weren't more models tested using lower degrees of freedom tested using SiteID as random intercept, as done for Year?

The reason for not showing the results of all models tested using SiteID as a random intercept was to reduce the dimensions of **TableS4**. From the first models tested, it became obvious that

using Year as random intercept led to lower AICc values (and a more parsimonious model) compared to those models using SiteID as random intercept.

We agree with the reviewer that either the data or further explanations should have been included in the text if those results were not shown. We have modified **Table S4** to include the models tested using SiteID as random intercept and lower degrees of freedom.

c) Are the random slopes and random intercepts correlated or uncorrelated? And how is this treated in the analysis?

The final model selected does not show any strong correlations between random slopes and random intercepts. Latitude is the only slope that shows a correlation with the random intercept ($r=0.47$), but without affecting model performance: when running the model after removing such correlation, the skills of the model remain unchanged, the AICc changes from 48236.12 to 48267.24, and it is still the most parsimonious model.

d) Why isn't year considered as a potential random slope effect? In fact, this would be a key element to test, since it would allow evaluating if potential differences in TRW between years due to random variations in growth, for example related with variations in C allocation to growth vs. reproduction, could contribute to some of the variability in the data.

We believe that the random effect structure that we used in the analysis encodes the assumptions that we make about how JSL influences vary across geographical location and year. The random slope connected to geographical variables allows for the explanatory variable to have a different effect depending on location. Year was always considered as a grouping factor because we considered it as non-independent data with repeated measurements.

That being said, and following the reviewer's suggestion, we have tested the inclusion of year as random slope along with the other random slopes tested in **Table S4** and considering site ID as a random intercept. The results show that the exclusion of year from the random components leads to a more parsimonious model (lower AICc) compared to the exclusion of any other variable considered as a random slope (see **Table A**). Therefore, including year as random intercept does not increase the amount of residual variability explained by the random effect component of the model.

Table A. Summary of the comparison of random effects including year for the linear mixed effects model (LMM) including the selected fixed effects (**Table S3**). ANOVA results comparing various LMMs that combine various random slopes and SiteID as random intercept. Abbreviations as in Table S2. lat: latitude; lon: longitude; ele: elevation; df: degrees of freedom. Bold font indicates the selected model based on the lowest AICc.

Random slope ($\beta\chi$)	df	Random intercept (β)	AICc
jslPC1, jslPC2, jslPC2y-1, lat, lon, ele, year	41	SiteID	56268.97
jslPC1, jslPC2, jslPC2y-1, lat, lon, ele	33	SiteID	54230.88
jslPC1, jslPC2, jslPC2y-1, lat, lon, year	33	SiteID	56049.82
jslPC1, jslPC2, jslPC2y-1, lat, ele, year	33	SiteID	56034.98
jslPC1, jslPC2, jslPC2y-1, lon, ele, year	33	SiteID	56017.02
jslPC1, jslPC2, lat, lon, ele, year	33	SiteID	56062.63
jslPC1, jslPC2y-1, lat, lon, ele, year	33	SiteID	56062.63
jslPC2, jslPC2y-1, lat, lon, ele, year	33	SiteID	55825.20

To improve clarity and rigor, I would recommend to rely more heavily on mathematical notation. For example, the authors can explicitly write the model equations in Tables S2-S4: $TRW = \beta_0 jslPC1 + \beta_1 jslPC2 + \dots$ and so on, or at least their right-hand sides, instead of simply the names of the variables.

We have changed the notation in the tables according to the reviewer's suggestion. In **Tables S2 and S3**, we have inserted an additional column with the model run at each step. Now, it is clear for every run, which predictors are included, and which are not (i.e., predictor excluded).

Regarding **Table S4**, since random slope and random intercept are listed in separate columns, we have included the mathematical notation at the top of every column. This way, we believe that the new table will be more legible and the information more accessible to a broader audience.

Thanks for explicitly showing the final model in Equation S1 now, it definitely improves clarity. But the notation is still unclear. What is γ_i and what does the i refer to? Is this TRW for each location? But then how are the random effects of elevation, latitude and longitude included in the model? The caption explains that $\mu_{1i|\text{lon,lat,ele}}$ refers to random slopes varying across geographical position (latitude, longitude and elevation), but the equation specifies that m depends on i ($\mu_{1i|\text{lon,lat,ele}}$). Which one is correct? Please provide full description of the different terms of the equation.

We have modified the equation and have fully described the different terms in the caption following the reviewer's suggestion.

Minor comments:

$jslPC1t$ and $jslPC2t$ are confusing acronyms, as one could think they relate to a trend. I suggest renaming them as $jslPC1y-1$ and $jslPC2y-1$ to explicitly show the lag.

We have modified the acronyms following the reviewer's suggestion.

Line 90-91: more correctly, it is 30% of modelled gross primary productivity. Carbon uptake would usually refer to NEE. Plus, the fluxes are derived from the DGVMs only.

We have replaced 'carbon uptake' by 'modelled gross primary productivity' as suggested by reviewer #1.

Line 120-122: It is still confusing that none of these two events is found in the analysis of extremes presented here. Although the authors addressed this in lines 444-455, but I do not find the explanation very convincing. In principle, the goal of using JSL is precisely to reduce the dimensionality of the drivers of TRW variability. Here is important to discuss the role of memory effects from soil drying in 2003, which has been shown to be a crucial factor to the development of the 2003 event (Fischer et al. 2007) and can likely explain why summer JSL isn't a good predictor then.

We agree with the reviewer that the role of memory effects from soil drying is a relevant or even a crucial factor determining the severity of a summer heatwave, as was the case in 2003. Although our paper focuses on the summer season, we acknowledge that it is relevant to mention the potential feedback of spring soil water deficit on the climate system early in the manuscript. Accordingly, we now mention the relationship between memory effects from soil

drying and severity of the 2003 summer heatwave more explicitly in the introduction (L126-128). We have additionally included in the Discussion (L467) an explicit mention to the potential contribution of soil moisture condition during the previous seasons on the occurrence of high-impact extreme events.

Line 661: “Starting from a saturated model, [...]” please explain what this means and clearly state what are the predictors of that model. To my understanding it is a model with all seven predictors listed in Table S2.

We have rewritten this section to clarify the steps that we followed (L684-704).

Table S2: my understanding is that the β , SE and p-values refer to the model with all seven predictors, while ΔAICc refers to the difference between that model and the ones excluding each predictor. This is not clear in the caption and should be stated explicitly for improved clarity and reproducibility.

We have clarified the captions according to the reviewer’s suggestion.

Table S3: is this now showing the coefficients for a model with all three predictors, and the ΔAICc the difference between AICc between that model and the ones with 2 predictors only? Please clarify in the caption.

We have rephrased the caption according to the reviewer’s suggestion.

Table S4: VIF is not shown, why?

The reason for not showing VIF is the large dimensions of the table. VIF values for each fixed and random effect for a table with the dimensions of **Table S4** will not only be large, but also not very informative since, as stated in the caption, variables displaying $\text{VIF} > 2$ were dropped from the model and there is no collinearity in the final selected model. However, we can provide such a table at the reviewer’s and/or editor’s discretion.

Fischer, E.M., Seneviratne, S.I., Vidale, P.L., Lüthi, D., Schär, C., 2007. Soil moisture–atmosphere interactions during the 2003 European summer heatwave. *J. Climate* 20, 5081–5099.

We have cited this reference in the revised manuscript.

Reviewer #2 (Remarks to the Author):

I thank the authors for their thoughtful and well-elaborated responses to my earlier suggestions and remarks on the previous version of the manuscript.

We thank reviewer #2 for the positive evaluations of the revised manuscript.

I have no further comments, except for a few very small things:

*the link to the code on figshare does not work;

The link to figshare is ready and contains the data and code. It will be activated after acceptance.

*I would be tempted to also include the “R code using existing functions” on the figshare repository. Also performing a linear mixed effect model requires specific choices, setting of parameters, random effects, error distribution, likelihood estimation, etc. and these are only reproducible with the full code.

We are confident that with the new information provided in this revised version of the manuscript (see reviewer 1's comments) and the source data and code that will be available via figshare, the modelling exercise performed will be fully reproducible.

I look forward to seeing this paper "in press".

Thank you!

Reviewer #3 (Remarks to the Author):

The authors have fully addressed my concerns. I have no further comments on the manuscript.

We thank reviewer#3 for the positive evaluation of the revised manuscript.

References

Barr, D. J., Levy, R., Scheepers, C., & Tily, H. J. (2013). Random effects structure for confirmatory hypothesis testing: Keep it maximal. *Journal of Memory and Language*, 68(3), 255–278. <https://doi.org/https://doi.org/10.1016/j.jml.2012.11.001>

Zuur, A. F., Ieno, E. N., & Elphick, C. S. (2010). A protocol for data exploration to avoid common statistical problems. *Methods in Ecology and Evolution*, 1(1), 3–14. <https://doi.org/10.1111/j.2041-210x.2009.00001.x>

Reviewers' Comments:

Reviewer #1:

Remarks to the Author:

I thank the authors for the thorough review of the MS. I believe this improved considerably the clarity of the methodological approach and therefore make it easier for a broader audience to appreciate the value of the study. I have no further comments and look forward to seeing the MS published.

REVIEWER COMMENTS

Reviewer #1 (Remarks to the Author):

I thank the authors for the thorough review of the MS. I believe this improved considerably the clarity of the methodological approach and therefore make it easier for a broader audience to appreciate the value of the study. I have no further comments and look forward to seeing the MS published.

We thank reviewer#1 for the positive evaluation of the revised manuscript.